# PTQD: Accurate Post-Training Quantization for Diffusion Models

**Yefei He[1]   Luping Liu[1]   Jing Liu[2]   Weijia Wu[1]   Hong Zhou[1†]   Bohan Zhuang[2†]**

[1]Zhejiang University, China
[2]ZIP Lab, Monash University, Australia

## Abstract

Diffusion models have recently dominated image synthesis and other related generative tasks. However, the iterative denoising process is expensive in computations at inference time, making diffusion models less practical for low-latency and scalable real-world applications. Post-training quantization of diffusion models can significantly reduce the model size and accelerate the sampling process without requiring any re-training. Nonetheless, applying existing post-training quantization methods directly to low-bit diffusion models can significantly impair the quality of generated samples. Specifically, for each denoising step, quantization noise leads to deviations in the estimated mean and mismatches with the predetermined variance schedule. Moreover, as the sampling process proceeds, the quantization noise may accumulate, resulting in a low signal-to-noise ratio (SNR) during the later denoising steps. To address these challenges, we propose a unified formulation for the quantization noise and diffusion perturbed noise in the quantized denoising process. Specifically, we first disentangle the quantization noise into its correlated and residual uncorrelated parts regarding its full-precision counterpart. The correlated part can be easily corrected by estimating the correlation coefficient. For the uncorrelated part, we subtract the bias from the quantized results to correct the mean deviation and calibrate the denoising variance schedule to absorb the excess variance resulting from quantization. Moreover, we introduce a mixed-precision scheme for selecting the optimal bitwidth for each denoising step, which prioritizes lower bitwidths to expedite early denoising steps, while ensuring that higher bitwidths maintain a high signal-to-noise ratio (SNR) in the later steps. Extensive experiments demonstrate that our method outperforms previous post-training quantized diffusion models in generating high-quality samples, with only a $0.06$ increase in FID score compared to full-precision LDM-4 on ImageNet $256 \times 256$, while saving $19.9\times$ bit operations. Code is available at https://github.com/ziplab/PTQD.

## 1  Introduction

Diffusion models have demonstrated remarkable ability in generating high-quality samples in multiple fields [10, 5, 58, 18, 38, 30, 13, 52, 8, 49]. Compared to generative adversarial networks (GANs) [15] and variational autoencoders (VAEs) [27], diffusion models do not face the issue of mode collapse and posterior collapse, thus training is more stable. Nonetheless, the application of diffusion models is limited by two major bottlenecks. Firstly, diffusion models typically require hundreds of denoising steps to generate high-quality samples, making the process significantly slower than that of GANs. To address this, many studies [50, 36, 2, 28, 34] have proposed advanced training-free sampler to reduce the number of denoising iterations. Among them, a recent study DPM-solver [36] curtails the denoising process to ten steps by analytically computing the linear part of the diffusion ordinary

---

[†]Corresponding author. Email: `zhouh@mail.bme.zju.edu.cn`, `bohan.zhuang@gmail.com`

37th Conference on Neural Information Processing Systems (NeurIPS 2023).


Q-Diffusion (W4A8)       PTQD (W4A8)       Full Precision


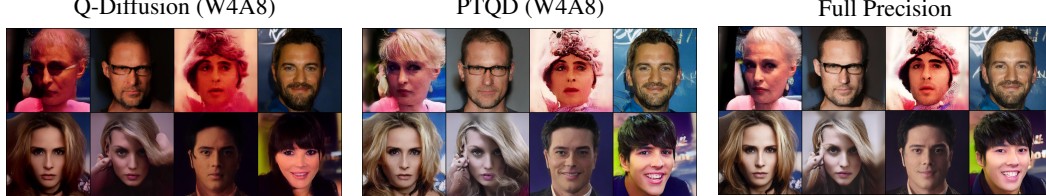

Figure 1: The comparisons of samples generated by Q-Diffusion [31], PTQD and full-precision LDM-4 [45] on CelebA-HQ $256 \times 256$ dataset. Here, W$x$A$y$ indicates the weights are quantized to $x$-bit while the activations are quantized to $y$-bit.

differential equations (ODEs). Nevertheless, diffusion models with these fast samplers are not yet ready for real-time applications. For instance, even when executed on a high-performance platform such as the RTX 3090, Stable Diffusion [45] with the DPM-Solver [36] sampler still takes over a second to generate a $512 \times 512$ image. Second, the application of diffusion models on various devices is constrained by the massive parameters and computational complexity. To illustrate, executing Stable Diffusion [45] requires 16GB of running memory and GPUs with over 10GB of VRAM, which is infeasible for most consumer-grade PCs, not to mention resource-constrained edge devices.

Model quantization, which employs lower numerical bitwidth to represent weights and activations, has been widely studied to reduce memory footprint and computational complexity. For instance, employing 8-bit models can result in a significant speed-up of $2.2\times$ compared to floating-point models on ARM CPUs [21]. Adopting 4-bit quantization can further deliver a throughput increase of up to $59\%$ compared to 8-bit quantization [3]. To facilitate the quantization process without the need for re-training, post-training quantization (PTQ) has emerged as a widely used technique, which is highly practical and easy to implement. While PTQ on traditional models have been widely studied [40, 32, 20, 33, 57], its application on diffusion models incurs two new challenges at the fundamental level. First, with the noise prediction network quantized, its quantization noise inevitably introduces bias in the estimated mean and brings additional variance that collides with the predetermined variance schedule in each denoising step. Additionally, the quantization noise accumulates as the iterative sampling process progresses, leading to a significant drop in the signal-to-noise ratio (SNR) of the noise prediction network in the later denoising steps. This diminished SNR severely impedes the denoising capability, resulting in a noticeable degradation in the quality of the generated images.

To tackle the aforementioned challenges, we present PTQD, a novel post-training quantization framework for diffusion models. To address the mean deviation and additional variance in each denoising step, we model the quantization noise by disentangling it into its correlated and residual uncorrelated parts regarding its full-precision counterpart, and designs separate correction methods for them. By estimating the correlation coefficient, the correlated part can be easily rectified. For the residual uncorrelated part, we subtract the bias from the estimated mean and propose variance schedule calibration, which absorbs the additional variance into the diffusion perturbed noise. To overcome the issue of low SNR that diminishes denoising capability in later denoising steps, we introduce a step-aware mixed precision scheme, which adaptively allocates different bitwidths for synonymous steps to maintain a high SNR for the denoising process.

In summary, our contributions are as follows:

- We present PTQD, a novel post-training quantization framework for diffusion models, which provides a unified formulation for quantization noise and diffusion perturbed noise.

- We disentangle the quantization noise into correlated and uncorrelated parts regarding its full-precision counterpart. Then we correct the correlated part by estimating the correlation coefficient, and propose variance schedule calibration to rectify the residual uncorrelated part.

- We introduce a step-aware mixed precision scheme, which dynamically selects the appropriate bitwidths for synonymous steps, preserving SNR throughout the denoising process.

- Our extensive experiments demonstrate that our method reaches a new state-of-the-art performance for post-training quantization of diffusion models.

## 2 Related Work

**Efficient diffusion models.** While diffusion models can produce high-quality samples, their slow generation speed hinders their large-scale applications in downstream tasks. To explore efficient diffusion models, many methods have been proposed to expedite the sampling process. These methods can be classified into two categories: methods that necessitate re-training and advanced samplers for pre-trained models that do not require training. The first category of methods comprises knowledge distillation [37, 47], diffusion scheme learning [7, 12, 64, 39], noise scale learning [26, 43], and sample trajectory learning [55, 29]. Although these methods can accelerate sampling, re-training a diffusion model can be resource-intensive and time-consuming. On the other hand, the second category of methods designs advanced samplers directly on pre-trained diffusion models, eliminating the need for re-training. The primary methods in this category are implicit sampler [50, 28, 63, 56], analytical trajectory estimation [2, 1], and differential equation (DE) solvers such as customized SDE [51, 23, 25] and ODE [36, 34, 62]. Although these methods can reduce the sampling iterations, the diffusion model's massive parameters and computational complexity restrict their use to high-performance platforms. Conversely, our proposed low-bit diffusion model can significantly reduce the model's computational complexity while speeding up the sampling and reducing the demand for hardware computing resources in a training-free manner.

**Model quantization.** Quantization is a dominant technique to save memory costs and speed up computation. It can be divided into two categories: quantization-aware training (QAT) [14, 35, 22, 65, 61] and post-training quantization (PTQ) [32, 40, 20, 57, 33]. QAT involves simulating quantization during training to achieve good performance with lower precision, but it requires substantial time, computational resources, and access to the original dataset. In contrast, PTQ does not require fine-tuning and only needs a small amount of unlabeled data to calibrate. Recent studies have pushed the limits of PTQ to 4-bit on traditional models by using new rounding strategies [40], layer-wise calibration [20, 53], and second-order statistics [32, 57]. Additionally, mixed precision (MP) [19, 11, 4, 59, 6] allows a part of the model to be represented by lower bitwidths to accelerate inference. Common criteria for determining quantization bitwidths include Hessian spectrum [11, 6] or Pareto frontier [4]. In contrast, we propose a novel mixed-precision scheme for diffusion models that adapts different bitwidths for synonymous denoising steps.

Until now, there have been few studies specifically focusing on quantizing a pre-trained diffusion model without re-training. PTQ4DM [48] is the first attempt to quantize diffusion models to 8-bit, but its experiments are limited to small datasets and low resolution. Q-Diffusion [31] applies advanced PTQ techniques proposed by BRECQ [32] to improve performance and evaluate it on a wider range of datasets. Our paper aims to analyze systematically the quantization effect on diffusion models and establish a unified framework for accurate post-training diffusion quantization.

## 3 Preliminaries

### 3.1 Diffusion Models

Diffusion models [50, 17] gradually apply Gaussian noise to real data $\mathbf{x}_0$ in the forward process and learn a reverse process to denoise and generate high-quality images. For DDPMs [17], the forward process is a Markov chain, which can be formulated as:

$$q(\mathbf{x}_{1:T}|\mathbf{x}_0) = \prod_{t=1}^{T} q(\mathbf{x}_t|\mathbf{x}_{t-1}), \qquad q(\mathbf{x}_t|\mathbf{x}_{t-1}) = \mathcal{N}(\mathbf{x}_t; \sqrt{\alpha_t}\mathbf{x}_{t-1}, \beta_t\mathbf{I}) \tag{1}$$

where $\alpha_t, \beta_t$ are hyperparameteres and $\beta_t = 1 - \alpha_t$.

In the reverse process, since directly estimating the real distribution of $q(\mathbf{x}_{t-1}|\mathbf{x}_t)$ is intractable, diffusion models approximate it via variational inference by learning a Gaussian distribution $p_\theta(\mathbf{x}_{t-1}|\mathbf{x}_t) = \mathcal{N}(\mathbf{x}_{t-1}; \boldsymbol{\mu}_\theta(\mathbf{x}_t, t), \boldsymbol{\Sigma}_\theta(\mathbf{x}_t, t))$ and reparameterize its mean by a noise prediction network $\boldsymbol{\epsilon}_\theta(\mathbf{x}_t, t)$:

$$\boldsymbol{\mu}_\theta(\mathbf{x}_t, t) = \frac{1}{\sqrt{\alpha_t}}\left(\mathbf{x}_t - \frac{\beta_t}{\sqrt{1-\bar{\alpha}_t}}\boldsymbol{\epsilon}_\theta(\mathbf{x}_t, t)\right) \tag{2}$$

where $\bar{\alpha}_t = \prod_{s=1}^{t} \alpha_s$. The variance $\boldsymbol{\Sigma}_\theta(\mathbf{x}_t, t)$ can either be reparameterized or fixed to a constant schedule $\sigma_t$. When it uses a constant schedule, the sampling of $\mathbf{x}_{t-1}$ can be formulated as:

$$\mathbf{x}_{t-1} = \frac{1}{\sqrt{\alpha_t}} \left( \mathbf{x}_t - \frac{\beta_t}{\sqrt{1 - \bar{\alpha}_t}} \boldsymbol{\epsilon}_\theta(\mathbf{x}_t, t) \right) + \sigma_t \mathbf{z}, \text{ where } \mathbf{z} \sim \mathcal{N}(\mathbf{0}, \mathbf{I}). \tag{3}$$

Our method focuses on post-training quantization of diffusion models without the need of training. Instead, we use pre-trained diffusion models and inherit their hyperparameters and variance schedules for inference. Although the derivations presented in this paper are based on DDPM, they can be readily extended to other fast sampling methods, such as DDIM [50]. Additional information can be found in the supplementary material.

## 3.2 Model Quantization

We use uniform quantization in our study and all the experiments. For uniform quantization, given a floating-point vector $\mathbf{x}$, the target bitwidth $b$, the quantization process can be defined as:

$$\hat{\mathbf{x}} = \Delta \cdot \left( \text{clip}(\lfloor \frac{\mathbf{x}}{\Delta} \rceil + Z, 0, 2^b - 1) - Z \right), \tag{4}$$

where $\lfloor \cdot \rceil$ is the round operation, $\Delta = \frac{\max(\mathbf{x}) - \min(\mathbf{x})}{2^b - 1}$ and $Z = -\lfloor \frac{\min(\mathbf{x})}{\Delta} \rceil$.

To ensure clarity and consistency, we introduce notation to define the variables used in the paper. Let $X$ be a tensor (weights or activations) in the full-precision model, the result after normalization layers is denoted as $\overline{X}$. The corresponding tensor of the quantized model is represented as $\hat{X}$. The quantization noise is depicted by $\Delta_X$, which is the difference between $\hat{X}$ and $X$.

# 4 Method

Model quantization discretizes the weights and activations, which will inevitably introduce quantization noise into the result. As per Eq. (3), during the reverse process of the quantized diffusion model, the sampling of $\mathbf{x}_{t-1}$ can be expressed as:

$$\mathbf{x}_{t-1} = \frac{1}{\sqrt{\alpha_t}} \left( \mathbf{x}_t - \frac{\beta_t}{\sqrt{1 - \bar{\alpha}_t}} \hat{\boldsymbol{\epsilon}}_\theta(\mathbf{x}_t, t) \right) + \sigma_t \mathbf{z} \tag{5}$$

$$= \frac{1}{\sqrt{\alpha_t}} \left( \mathbf{x}_t - \frac{\beta_t}{\sqrt{1 - \bar{\alpha}_t}} \left( \boldsymbol{\epsilon}_\theta(\mathbf{x}_t, t) + \Delta_{\boldsymbol{\epsilon}_\theta(\mathbf{x}_t, t)} \right) \right) + \sigma_t \mathbf{z}.$$

Here, $\hat{\boldsymbol{\epsilon}}_\theta(\mathbf{x}_t, t)$ is the output of the quantized noise prediction network and $\Delta_{\boldsymbol{\epsilon}_\theta(\mathbf{x}_t, t)}$ refers to the quantization noise. The additional quantization noise will inevitably alter the mean and variance of $\mathbf{x}_{t-1}$, decreasing the signal-to-noise ratio (SNR) and adversely affecting the quality of the generated samples. Therefore, to mitigate the impact of quantization, it is necessary to correct the mean and variance to restore the SNR at each step of the reverse process.

## 4.1 Correlation Disentanglement

We begin by making an assumption that a correlation exists between the quantization noise and the result of the full-precision noise prediction network. While other factors, such as nonlinear operations, may contribute to this correlation, *Proposition 1* demonstrates that normalization layers are responsible for a part of it.

**Proposition 1.** *Given $Y$ and $\hat{Y}$ as inputs to a normalization layer in a full-precision model and its quantized version, where the quantization noise $\Delta_Y = \hat{Y} - Y$ is initially uncorrelated with $Y$, a correlation between the quantization noise and the output of the full-precision model after the normalization layer will exist.*

The proof is based on the fact that the mean and variance of $\hat{Y}$ will differ from that of $Y$ (depending on the specific quantization scheme). Therefore, the quantization noise after normalization layer can be expressed as :

$$\Delta_{\overline{Y}} = \frac{\hat{Y} - \mu_{\hat{Y}}}{\sigma_{\hat{Y}}} - \frac{Y - \mu_Y}{\sigma_Y} = \frac{\sigma_Y \Delta_Y - (\sigma_{\hat{Y}} - \sigma_Y)Y + \sigma_{\hat{Y}} \mu_Y - \sigma_Y \mu_{\hat{Y}}}{\sigma_{\hat{Y}} \sigma_Y}. \tag{6}$$

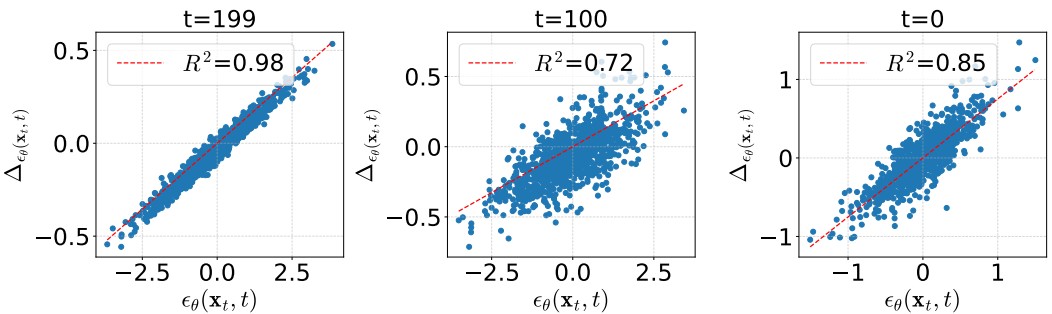

Figure 2: The correlation between the quantization noise (Y-axis) and the output of the full-precision noise prediction network (X-axis). Each data point on the plot corresponds to specific entries within these vectors. Data were collected by generating samples with 4-bit LDM-8 [45] for 200 steps on LSUN-Churches [60].

Here, we omit the affine transform parameters in normalization layers for simplicity. It can be observed from Eq. (6) that the second term in the numerator is related to $Y$, while the other three terms are uncorrelated. Therefore, after normalization layers, the quantization noise $\Delta_{\overline{Y}}$ will be correlated with $Y$.

The empirical observation illustrated in Figure 2 confirms a strong correlation between the quantization noise and the output of the full-precision noise prediction network, which further verifies our assumption. Based on the assumption and observation, the quantization noise of the quantized noise prediction network can be disentangled into two parts:

$$\Delta_{\boldsymbol{\epsilon}_\theta(\mathbf{x}_t,t)} = k\boldsymbol{\epsilon}_\theta(\mathbf{x}_t,t) + \Delta'_{\boldsymbol{\epsilon}_\theta(\mathbf{x}_t,t)}. \tag{7}$$

The first part, denoted by $k\boldsymbol{\epsilon}_\theta(\mathbf{x}_t,t)$, is linearly related to $\boldsymbol{\epsilon}_\theta(\mathbf{x}_t,t)$. The second part, expressed by $\Delta'_{\boldsymbol{\epsilon}_\theta(\mathbf{x}_t,t)}$, represents the residual component of the quantization noise, and is assumed to be uncorrelated with $\boldsymbol{\epsilon}_\theta(\mathbf{x}_t,t)$. Here, $k$ is the correlation coefficient, which can be estimated by applying linear regression on the quantization noise $\Delta_{\boldsymbol{\epsilon}_\theta(\mathbf{x}_t,t)}$ and the original value $\boldsymbol{\epsilon}_\theta(\mathbf{x}_t,t)$. Details can be found in Section 5.1.

With the disentanglement presented in Eq. (7), the sampling of $\mathbf{x}_{t-1}$ can be further expressed as:

$$\mathbf{x}_{t-1} = \frac{1}{\sqrt{\alpha_t}} \left( \mathbf{x}_t - \frac{\beta_t}{\sqrt{1-\bar{\alpha}_t}} \left( \boldsymbol{\epsilon}_\theta(\mathbf{x}_t,t) + \Delta_{\boldsymbol{\epsilon}_\theta(\mathbf{x}_t,t)} \right) \right) + \sigma_t \mathbf{z} \tag{8}$$

$$= \frac{1}{\sqrt{\alpha_t}} \left( \mathbf{x}_t - \frac{\beta_t}{\sqrt{1-\bar{\alpha}_t}} \left( (1+k)\,\boldsymbol{\epsilon}_\theta(\mathbf{x}_t,t) + \Delta'_{\boldsymbol{\epsilon}_\theta(\mathbf{x}_t,t)} \right) \right) + \sigma_t \mathbf{z}.$$

Consequently, the bias and additional variance arise from both the correlated and uncorrelated parts of quantization noise. In the following section, we will provide a detailed explanation of how these two parts of quantization noise can be separately corrected.

## 4.2 Quantization Noise Correction

### 4.2.1 Correlated Noise Correction

Based on Eq. (8), the correlated part of the quantization noise can be rectified by dividing the output of the quantized noise prediction network $\hat{\boldsymbol{\epsilon}}_\theta(\mathbf{x}_t,t)$ by $1+k$, resulting in:

$$\mathbf{x}_{t-1} = \frac{1}{\sqrt{\alpha_t}} \left( \mathbf{x}_t - \frac{\beta_t}{\sqrt{1-\bar{\alpha}_t}}\boldsymbol{\epsilon}_\theta(\mathbf{x}_t,t) \right) + \sigma_t \mathbf{z} - \frac{\beta_t}{\sqrt{\alpha_t}\sqrt{1-\bar{\alpha}_t}(1+k)}\Delta'_{\boldsymbol{\epsilon}_\theta(\mathbf{x}_t,t)}. \tag{9}$$

Consequently, only the uncorrelated quantization noise remains. Moreover, for values of $k \geq 0$, it can be deduced that the mean and variance of the uncorrelated quantization noise are diminished by $\frac{1}{1+k}$. In practice, we enforce the non-negativity of $k$, and reset it to zero if it is negative. In the following, we will explain how to handle the uncorrelated quantization noise that persists in Eq. (9).

#### 4.2.2 Uncorrelated Noise Correction

The presence of uncorrelated quantization noise introduces additional variance at each step, resulting in a total variance that exceeds the scheduled value $\sigma_t^2$. To address this, we propose to calibrate the variance schedule for quantized diffusion models, which is denoted as $\sigma_t^{'2}$ and smaller than the original schedule $\sigma_t^2$. To estimate $\sigma_t^{'2}$, we further make an assumption and model the uncorrelated quantization noise as a Gaussian distribution with a mean of $\mu_q$ and a variance of $\sigma_q^2$:

$$\Delta_{\boldsymbol{\epsilon}_\theta(\mathbf{x}_t,t)}^{'} \sim \mathcal{N}(\mu_q, \sigma_q). \tag{10}$$

To verify this assumption, we conduct statistical tests (refer to the supplementary material) and present the distribution of the uncorrelated quantization noise in Figure 3. The values of mean and variance can be estimated by generating samples with both quantized and full-precision diffusion models and collecting the statistics of the uncorrelated quantization noise. Following prior work [41], the mean deviation can be rectified through Bias Correction (BC), where we collect the channel-wise means of uncorrelated quantization noise and subtract them from the output of quantized noise prediction network. For the variance of the uncorrelated quantization noise, we propose Variance Schedule Calibration (VSC), where the uncorrelated quantization noise can be absorbed into Gaussian diffusion noise with the above assumption. By substituting the calibrated variance schedule $\sigma_t^{'2}$ into Eq. (9) while keeping the variance of each step unaltered, we can solve for the optimal variance schedule using the following approach:

$$\sigma_t^{'2} + \frac{\beta_t^2}{\alpha_t(1 - \bar{\alpha}_t)(1 + k)^2}\sigma_q^2 = \sigma_t^2, \tag{11}$$

$$\sigma_t^{'2} = \begin{cases} \sigma_t^2 - \frac{\beta_t^2}{\alpha_t(1 - \bar{\alpha}_t)(1 + k)^2}\sigma_q^2, & \text{if } \sigma_t^2 \geq \frac{\beta_t^2}{\alpha_t(1 - \bar{\alpha}_t)(1 + k)^2}\sigma_q^2 \\ 0, & \text{otherwise.} \end{cases} \tag{12}$$

It can be observed that if the additional variance of quantization noise is smaller than the noise hyperparameter $\sigma_t^2$, the increase in variance caused by quantization can be eliminated. According to Eq. (12), the coefficient for the variance of the quantization noise can be calculated as $\frac{\beta_t^2}{\alpha_t(1 - \bar{\alpha}_t)(1 + k)^2}$, which is generally small enough to ensure that the quantization noise can be fully absorbed, except for cases of deterministic sampling where $\sigma_t$ is zero. In this case, there is no analytical solution for $\sigma_t^{'2}$, and we use the optimal solution that is $\sigma_t^{'2} = 0$. Overall, the process quantization noise correction is summarized in Algorithm 1.

---

**Algorithm 1:** Quantization noise correction.

---
**Statistics collection before sampling:**
1) Quantize diffusion models with BRECQ [32] (or other PTQ methods);
2) Generate samples with both quantized and FP models and collect quantization noise;
3) Calculate the correlated coefficient $k$ based on Eq. (7), and the mean and variance of the uncorrelated quantization noise as per Eq. (10);
**Noise correction for each sampling step:**
4) Correct the correlated part of the quantization noise by dividing the output of the noise prediction network by $1 + k$;
5) Calibrate the variance schedule by Eq. (12) and subtract the channel-wise biases from the output of the quantized noise prediction network.

---

Although the proposed method can correct the mean deviation and the numerical value of the variance for each step, generating satisfactory samples with low-bit diffusion models remains challenging due to the low signal-to-noise ratio (SNR) of the quantized noise prediction network. In the next section, we will analyze this issue in detail.

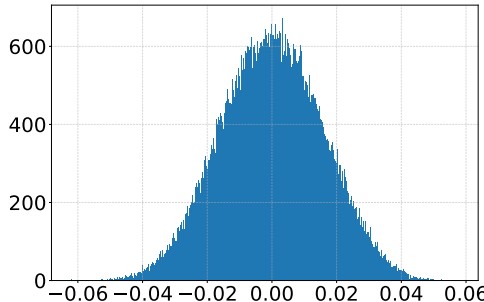
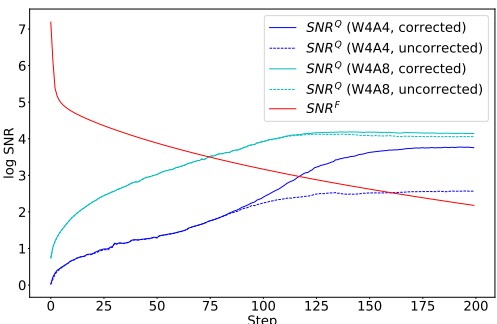

Figure 3: The distribution of uncorrelated quantization noise collected from W4A8 LDM-4 on LSUN-Bedrooms $256 \times 256$ dataset, where the x-axis represents the range of values and the y-axis is the frequency of values.

Figure 4: Comparison of the signal-to-noise-ratio (SNR) in each step of LDM-4 on LSUN-Bedrooms across various bitwidths.

### 4.3 Step-aware Mixed Precision

Given the output of the full-precision noise prediction network $\epsilon_\theta(\mathbf{x}_t, t)$ and corresponding quantization noise $\Delta_{\epsilon_\theta(\mathbf{x}_t, t)}$, we define the $\mathrm{SNR}^\mathrm{Q}$ of quantized noise prediction network by:

$$\mathrm{SNR}^\mathrm{Q}(t) = \frac{\|\epsilon_\theta(\mathbf{x}_t, t)\|_2}{\|\Delta_{\epsilon_\theta(\mathbf{x}_t, t)}\|_2}. \tag{13}$$

Figure 4 depicts the $\mathrm{SNR}^\mathrm{Q}$ with various bitwidths and correction methods. The figure reveals several insights: 1) $\mathrm{SNR}^\mathrm{Q}$ drops drastically as step $t$ decreases; 2) models with higher bitwidth exhibit larger $\mathrm{SNR}^\mathrm{Q}$; 3) the proposed correction methods yield clear $\mathrm{SNR}^\mathrm{Q}$ improvements, especially for large steps. The first observation highlights the challenge of generating high-quality samples using low-bit diffusion models. In particular, as $t$ approaches zero, the $\mathrm{SNR}^\mathrm{Q}$ of W4A4 diffusion models diminishes and approaches unity, implying that the magnitude of quantization noise is even comparable to the original result of the noise prediction network. To enable low-bit diffusion models while maintaining good generation performance, we propose a novel approach called Step-aware Mixed Precision, which involves setting different bitwidths for synonymous steps to keep $\mathrm{SNR}^\mathrm{Q}$ within a reasonable range across all steps.

Specifically, the bitwidth of weights is fixed and shared across different denoising steps, which eliminates the need to store and reload multiple model state files during the sampling process. As a result, we only adjust the bitwidth of activations. Formally, we predefine a set of bitwidths $B = \{b_1, b_2, \ldots, b_n\}$ for activations and evaluate the $\mathrm{SNR}^\mathrm{Q}$ under each bitwidth. To establish a benchmark for $\mathrm{SNR}^\mathrm{Q}$, we follow prior studies [36, 26] and introduce $\mathrm{SNR}^\mathrm{F}$ based on the forward process, which denotes the degree of data noise at each step:

$$\mathrm{SNR}^\mathrm{F}(t) = \alpha_t^2 / \sigma_t^2. \tag{14}$$

Figure 4 illustrates $\mathrm{SNR}^\mathrm{F}(t)$, which decreases strictly with respect to steps $t$. To determine the optimal bitwidth for each step $t$, we compare the $\mathrm{SNR}^\mathrm{Q}$ of each bitwidth with $\mathrm{SNR}^\mathrm{F}$, and select the minimum bitwidth $b_\mathrm{min}$ that satisfies:

$$\mathrm{SNR}^\mathrm{Q}_{b_\mathrm{min}}(t) > \mathrm{SNR}^\mathrm{F}(t). \tag{15}$$

If none of the bitwidths satisfies this condition, we utilize the maximum bitwidth in $B$ to achieve a higher SNR. In practice, models with different bitwidths are calibrated separately, with the calibration set collected from the corresponding steps.

# 5 Experiments

## 5.1 Implementation Details

**Datasets and quantization settings.** We conduct image synthesis experiments using latent diffusion models (LDM) [45] on three standard benchmarks: ImageNet[9], LSUN-Bedrooms, and LSUN-Churches [60], each with a resolution of $256 \times 256$. All experimental configurations, including the number of steps, variance schedule (denoted by $eta$ in the following), and classifier-free guidance scale, follow the official implementation [45]. For low-bit quantization, we use the PTQ method proposed in BRECQ [32] and AdaRound [40], which is congruent with Q-Diffusion [31]. For 8-bit quantization on ImageNet, we only use a naive PTQ method proposed by TensorRT [44], which is simple and fast. The input and output layers in the model are fixed to 8-bit, while all other convolutional and linear layers are quantized to the target bitwidth. In mixed precision experiments, we fix the weights to 4-bit and use Eq. (15) to determine the bitwidth of activations over uncorrected quantized diffusion models with a bitwidth set of $\{4, 8\}$. Details of bitwidth allocation can be found in the supplementary material.

**Evaluation metrics.** For each experiment, we report the widely adopted Frechet Inception Distance (FID) [16] and sFID [42] to evaluate the performance. For ImageNet experiments, we additionally report Inception Score (IS) [46] for reference. To ensure consistency in the reported outcomes, including those of the baseline methods, all results are obtained by our implementation. We sample 50,000 images and evaluate them with ADM's TensorFlow evaluation suite [10]. To quantify the computational efficiency, we measure Bit Operations (BOPs) for a single forward pass of the diffusion model using the equation $\text{BOPs} = \text{MACs} \cdot \text{b}_\text{w} \cdot \text{b}_\text{a}$, where $\text{MACs}$ denotes Multiply-And-Accumulate operations, and $\text{b}_\text{w}$ and $\text{b}_\text{a}$ represent the bitwidth of weights and activations, respectively, following [54].

**Statistics collection.** Before implementing our method, three statistics need to be collected: the correlation coefficient, denoted as $k$ in Eq. (7), and the mean and variance of the uncorrelated quantization noise, as depicted in Eq. (10). To obtain these statistics, we generate 1024 samples using both quantized and full-precision diffusion models, store the quantization noise at each step, and then calculate the required statistics.

## 5.2 Ablation Study

As shown in Table 1, we conduct ablation experiments on ImageNet $256 \times 256$ dataset over LDM-4 model, to demonstrate the effectiveness of the proposed techniques. These techniques include Correlated Noise Correction (CNC) for addressing the correlated quantization noise, as well as Bias Correction (BC) and Variance Schedule Calibration (VSC) for correcting the residual uncorrelated quantization noise. By employing Correlated Noise Correction, we achieved a 0.48 reduction in FID and a 6.55 decrease in sFID. The considerable reduction in sFID suggests that the generated images possess more intricate spatial details than those generated using the baseline method, and that the correlated portions significantly contribute to the quantization noise. With the proposed Variance Schedule Calibration, the additional variance of uncorrelated quantization noise can be absorbed, achieving a reduction of 0.2 in FID and 0.11 in sFID. By further introducing Bias Correction that effectively corrects the mean deviation caused by quantization noise, our proposed PTQD achieved an FID of 6.44 and an sFID of 8.43, with only a 1.33 increase in sFID under the W4A4/W4A8 mixed precision setting. These results demonstrate the efficacy of the proposed techniques in achieving accurate post-training quantization of diffusion models.

Additional ablation experiments can be found in the supplementary material.

## 5.3 Main Results

### 5.3.1 Class-conditional Generation

In this section, we evaluate the performance of class-conditional image generation on $256 \times 256$ ImageNet dataset, as presented in Table 2. By utilizing the Naive PTQ method [44] and quantizing to 8-bit, diffusion models can achieve a notable $12.39\times$ reduction in bit operations, while experiencing minimal increases in FID/sFID. With the aggressive W4A8 bitwidth setting, our method effectively narrows the FID gap to a mere 0.06 with 250 generation steps. In this setting, the model size is

Table 1: The effect of different components proposed in the paper. Here, MP denotes the proposed step-aware mixed precision scheme.

| Models | Method | Bitwidth (W/A) | FID↓ | sFID↓ |
|---|---|---|---|---|
| LDM-4 (steps = 250 eta = 1.0 scale = 1.5) | Q-Diffusion | MP | 9.97 | 18.23 |
| | + CNC | MP | 9.49 | 11.68 |
| | + CNC + VSC | MP | 9.29 | 11.57 |
| | PTQD (CNC + VSC + BC) | MP | 6.44 | 8.43 |
| | FP | 32/32 | 5.05 | 7.10 |

compressed by $6.83\times$ and the bit operations can be reduced by a remarkable $19.96\times$. In experiments utilizing mixed precision with W4A4 and W4A8 bitwidths, previous methods encounter difficulties in mitigating substantial quantization noises caused by low-bit quantization. For instance, in the first set of experiments with a 20-step generation process, Q-Diffusion [31] obtains FID and sFID scores as high as 116.61 and 172.99, respectively, indicating difficulties in handling low-bit diffusion models with fewer generation steps. While our method cannot calibrate the variance schedule due to a zero value for the hyperparameter $eta$, it still achieves an exceptionally low FID score of 7.75, demonstrating effective rectification of the correlated quantization noise and mean deviation. The second set of mixed precision experiments also yielded similar results, with our method reducing FID and sFID scores by 3.53 and 9.80, respectively.

Table 2: Performance comparisons of class-conditional image generation on ImageNet $256 \times 256$.

| Model | Method | Bitwidth (W/A) | Model Size (MB) | BOPs (T) | BOP comp. ratio | IS↑ | FID↓ | sFID↓ |
|---|---|---|---|---|---|---|---|---|
| LDM-4 (steps = 20 eta = 0.0 scale = 3.0) | FP | 32/32 | 1603.35 | 102.21 | - | 225.16 | 12.45 | 7.85 |
| | Naive PTQ | 8/8 | 430.06 | 8.25 | 12.39× | 152.91 | 12.14 | 8.43 |
| | Ours | 8/8 | 430.06 | 8.25 | 12.39× | **153.92** | **11.94** | **8.03** |
| | Q-Diffusion | 4/8 | 234.51 | 5.12 | 19.96× | 212.52 | 10.63 | 14.80 |
| | Ours | 4/8 | 234.51 | 5.12 | 19.96× | **214.73** | **10.40** | **12.68** |
| | Q-Diffusion | MP | 234.51 | 4.73 | 21.61× | 7.86 | 116.61 | 172.99 |
| | Ours | MP | 234.51 | 4.73 | 21.61× | **175.19** | **7.75** | **18.78** |
| LDM-4 (steps = 250 eta = 1.0 scale = 1.5) | FP | 32/32 | 1603.35 | 102.21 | - | 185.04 | 5.05 | 7.10 |
| | Naive PTQ | 8/8 | 430.06 | 8.25 | 12.39× | 180.56 | 4.06 | 5.91 |
| | Ours | 8/8 | 430.06 | 8.25 | 12.39× | **180.83** | **4.02** | **5.81** |
| | Q-Diffusion | 4/8 | 234.51 | 5.12 | 19.96× | 148.74 | 5.37 | 9.56 |
| | Ours | 4/8 | 234.51 | 5.12 | 19.96× | **149.74** | **5.11** | **8.49** |
| | Q-Diffusion | MP | 234.51 | 4.81 | 21.25× | 121.10 | 9.97 | 18.23 |
| | Ours | MP | 234.51 | 4.81 | 21.25× | **126.26** | **6.44** | **8.43** |

## 5.4 Unconditional Generation

In this section, we present a comprehensive evaluation of our approach on LSUN-Bedrooms and LSUN-Churches [60] datasets for unconditional image generation. As shown in Table 3, our method consistently narrows the performance gap between quantized and full-precision diffusion models. Notably, our proposed method allows for compression of diffusion models to 8-bit with minimal performance degradation, resulting in a mere 0.1 increase in FID on the LSUN-Churches dataset. With the W4A8 bitwidth setting, our method reduces FID and sFID by notably 0.78 and 3.61 compared with Q-Diffusion [31] on LSUN-Bedrooms. Furthermore, Q-Diffusion fails to effectively denoise samples under the mixed precision setting on LSUN-Churches due to its low SNR. In this case, the hyperparameter $eta$ is set to zero, which prevents the use of Variance Schedule Calibration. Despite relying solely on Correlated Noise Correction and Bias Correction, our approach remarkably enhances the quality of the generated images, as demonstrated by a substantial reduction in the FID score from 218.59 to 17.99. This notable improvement highlights the significant impact of the correlated part of quantization noise on the overall image quality, which can be effectively rectified by our method.

Additional evaluation results on CelebA-HQ dataset can be found in the supplementary material.

Table 3: Performance comparisons of unconditional image generation.

| LSUN-Bedrooms $256 \times 256$ LDM-4 (steps = 200, eta = 1.0) | | | | LSUN-Churches $256 \times 256$ LDM-8 (steps = 200, eta = 0.0) | | | |
|---|---|---|---|---|---|---|---|
| Method | Bitwidth (W/A) | FID↓ | sFID↓ | Method | Bitwidth (W/A) | FID↓ | sFID↓ |
| Full precision | 32/32 | 3.00 | 7.13 | Full precision | 32/32 | 6.30 | 18.24 |
| Q-Diffusion | 8/8 | 3.80 | 9.95 | Q-Diffusion | 8/8 | 6.94 | 18.93 |
| Ours | 8/8 | **3.75** | **9.89** | Ours | 8/8 | **6.40** | **18.34** |
| Q-Diffusion | 4/8 | 6.72 | 18.80 | Q-Diffusion | 4/8 | 7.80 | 19.97 |
| Ours | 4/8 | **5.94** | **15.16** | Ours | 4/8 | **7.33** | **19.40** |
| Q-Diffusion | MP | 5.75 | 12.79 | Q-Diffusion | MP | 218.59 | 312.86 |
| Ours | MP | **5.49** | **12.04** | Ours | MP | **17.99** | **37.34** |

## 5.5 Deployment Efficiency

We have measured the latency of matrix multiplication and convolution operations in quantized and full-precision diffusion models using an RTX3090 GPU, as shown in Table 4. Both floating-point and quantized operations are implemented with CUTLASS [24]. When both weights and activations are quantized to 8-bit, we observe a $2.03\times$ reduction in latency compared to its full-precision counterpart over LDM-4. Moreover, when weights and activations are quantized to 4-bit, the speedup further increases to $3.34\times$. The mixed-precision settings explored in our experiments strike a good balance between latency and model performance.

Table 4: Comparisons of time cost across various bitwidth configurations on ImageNet $256 \times 256$. Due to the current lack of a fast implementation for W4A8, we implement MP scheme with W8A8 and W4A4 kernels.

| Model | Bitwidth (W/A) | Model Size (MB) | FID↓ | sFID↓ | Time (s) |
|---|---|---|---|---|---|
| LDM-4 | 32/32 | 1603.35 | 5.05 | 7.10 | 5.46 |
| (steps=250 | 8/8 | 430.06 | 4.02 | 5.81 | 2.68 |
| eta=1.0 | MP | 234.51 | 6.44 | 8.43 | 2.45 |
| scale=1.5) | 4/4 | 234.51 | - | - | 1.63 |

## 6 Conclusion and Future Work

In this paper, we have proposed PTQD, a novel post-training quantization framework for diffusion models that unifies the formulation of quantization noise and diffusion perturbed noise. To start with, we have disentangled the quantization noise into correlated and residual uncorrelated parts relative to its full-precision counterpart. To reduce mean deviations and additional variance in each step, the correlated part can be easily corrected by estimating the correlation coefficient. For the uncorrelated part, we have proposed Variance Schedule Calibration to absorb its additional variance and Bias Correction to correct the mean deviations. Moreover, we have introduced Step-aware Mixed Precision to adaptively select the optimal bitwidth for each denoising step. By incorporating these techniques, our PTQD has achieved significant performance improvement over existing state-of-the-art post-training quantized diffusion models, with only a 0.06 FID increase compared to the full-precision LDM-4 on ImageNet $256 \times 256$ while saving $19.9\times$ bit-operations. In the future, we can further quantize other components within diffusion models, such as the text encoder and image decoder, to achieve higher compression ratios and accelerated performance. We may also extend PTQD to a wider range of generative tasks to assess its efficacy and generalizability.

**Limitations and Broader Impacts.** The proposed PTQD framework stands out for its high efficiency and energy-saving properties, which carry significant implications in reducing the carbon emissions attributed to the widespread deployment of diffusion models. However, similar to other deep generative models, PTQD has the potential to be utilized for producing counterfeit images and videos for malicious purposes.

**Acknowledgement** This work was supported by National Key Research and Development Program of China (2022YFC3602601).

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
