# Supplementary Material for PTQD: Accurate Post-Training Quantization for Diffusion Models

**Yefei He[1]  Luping Liu[1]  Jing Liu[2]  Weijia Wu[1]  Hong Zhou[1†]  Bohan Zhuang[2†]**

[1]Zhejiang University, China
[2]ZIP Lab, Monash University, Australia

We organize our supplementary material as follows:

- In section A, we provide a comprehensive explanation of extending PTQD to DDIM [10].
- In section B, we show the statistical analysis of quantization noise.
- In section C, we present additional experimental results.
- In section D, we provide additional visualization results on ImageNet and LSUN dataset.

## A  Extending PTQD to DDIM

DDIM [10] generalizes DDPMs [3] via a class of non-Markovian diffusion processes, which can greatly accelerate the sampling process. Briefly, when DDIM is quantized, the sampling of $\mathbf{x}_{t-1}$ can be expressed as:

$$\mathbf{x}_{t-1} = \sqrt{\alpha_{t-1}} \left( \frac{\mathbf{x}_t - \sqrt{1-\alpha_t}\hat{\boldsymbol{\epsilon}}_\theta(\mathbf{x}_t,t)}{\sqrt{\alpha_t}} \right) + \sqrt{1-\alpha_{t-1}-\sigma_t^2}\hat{\boldsymbol{\epsilon}}_\theta(\mathbf{x}_t,t) + \sigma_t\mathbf{z} \tag{A}$$

$$= \sqrt{\alpha_{t-1}} \left( \frac{\mathbf{x}_t - \sqrt{1-\alpha_t}(\boldsymbol{\epsilon}_\theta(\mathbf{x}_t,t) + \Delta_{\boldsymbol{\epsilon}_\theta(\mathbf{x}_t,t)})}{\sqrt{\alpha_t}} \right) + \sqrt{1-\alpha_{t-1}-\sigma_t^2}(\boldsymbol{\epsilon}_\theta(\mathbf{x}_t,t) + \Delta_{\boldsymbol{\epsilon}_\theta(\mathbf{x}_t,t)}) + \sigma_t\mathbf{z}$$

where $\hat{\boldsymbol{\epsilon}}_\theta(\mathbf{x}_t,t)$ is the result of quantized noise prediction network and $\Delta_{\boldsymbol{\epsilon}_\theta(\mathbf{x}_t,t)}$ is the quantization noise.

Firstly, we disentangle the quantization noise to its correlated and residual uncorrelated part, which is same as in DDPM [3]:

$$\Delta_{\boldsymbol{\epsilon}_\theta(\mathbf{x}_t,t)} = k\boldsymbol{\epsilon}_\theta(\mathbf{x}_t,t) + \Delta'_{\boldsymbol{\epsilon}_\theta(\mathbf{x}_t,t)}. \tag{B}$$

Then we can reformulate Eq. (A) as

$$\mathbf{x}_{t-1} = \sqrt{\alpha_{t-1}} \left( \frac{\mathbf{x}_t - \sqrt{1-\alpha_t}\left((1+k)\boldsymbol{\epsilon}_\theta(\mathbf{x}_t,t) + \Delta'_{\boldsymbol{\epsilon}_\theta(\mathbf{x}_t,t)}\right)}{\sqrt{\alpha_t}} \right)$$
$$+ \sqrt{1-\alpha_{t-1}-\sigma_t^2}\left((1+k)\boldsymbol{\epsilon}_\theta(\mathbf{x}_t,t) + \Delta'_{\boldsymbol{\epsilon}_\theta(\mathbf{x}_t,t)}\right) + \sigma_t\mathbf{z}. \tag{C}$$

By estimating the correlation coefficient $k$, the correlated part can be corrected by dividing the output of the quantized noise prediction network $\hat{\boldsymbol{\epsilon}}_\theta(\mathbf{x}_t,t)$ by $1+k$:

$$\mathbf{x}_{t-1} = \sqrt{\alpha_{t-1}} \left( \frac{\mathbf{x}_t - \sqrt{1-\alpha_t}\boldsymbol{\epsilon}_\theta(\mathbf{x}_t,t)}{\sqrt{\alpha_t}} \right) + \sqrt{1-\alpha_{t-1}-\sigma_t^2}\boldsymbol{\epsilon}_\theta(\mathbf{x}_t,t)$$
$$+ (\frac{\sqrt{1-\alpha_{t-1}-\sigma_t^2}}{1+k} - \frac{\sqrt{\alpha_{t-1}}\sqrt{1-\alpha_t}}{(1+k)\sqrt{\alpha_t}})\Delta'_{\boldsymbol{\epsilon}_\theta(\mathbf{x}_t,t)} + \sigma_t\mathbf{z}. \tag{D}$$

---

[†]Corresponding author. Email: `zhouh@mail.bme.zju.edu.cn`, `bohan.zhuang@gmail.com`

37th Conference on Neural Information Processing Systems (NeurIPS 2023).

Then we calibrate the variance schedule, denoted as $\sigma_t'$, to absorb the excess variance of residual quantization noise, which is depicted by $\sigma_q^2$. Let $\lambda_t = \frac{\sqrt{1-\alpha_{t-1}-\sigma_t^2}}{1+k} - \frac{\sqrt{\alpha_{t-1}}\sqrt{1-\alpha_t}}{(1+k)\sqrt{\alpha_t}}$, we have:

$$\lambda_t^2 \sigma_q^2 + \sigma_t'^2 = \sigma_t^2, \tag{E}$$

$$\sigma_t'^2 = \begin{cases} \sigma_t^2 - \lambda_t^2 \sigma_q^2, & \text{if } \sigma_t^2 \geq \lambda_t^2 \sigma_q^2 \\ 0, & \text{otherwise.} \end{cases} \tag{F}$$

## B  Statistical analysis

**Distribution of residual quantization noise.** We first perform statistical tests to verify if the residual quantization noise adheres to a Gaussian distribution. To accomplish this, we employ the significance test *scipy.stats.normaltest* provided by Scipy [12]. This test is based on D'Agostino and Pearson's test [2, 1], with the null hypothesis proposing that the sample comes from a normal distribution. The outcomes are illustrated in Figure A, and they reveal that, with a significance level of $0.01$, the null hypothesis cannot be rejected at any step, thus substantiating our assumption. In Figure B, we present the variance of the residual uncorrelated quantization noise. It can be observed that as the quantization bitwidth decreases, the variance of the quantization noise increases accordingly. Nonetheless, the coefficient associated with this variance is relatively small, allowing for its effective absorption into the calibrated diffusion variance schedule. Figure C illustrates the bias on the estimated mean introduced by the residual quantization noise. Notably, this bias exhibits significant variations across different channels, emphasizing the necessity for distinct correction procedures for each channel.

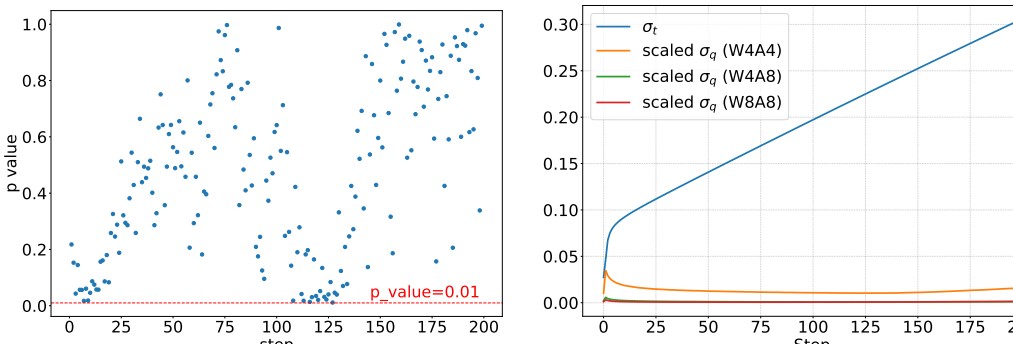

Figure A: The result of normal test for residual quantization noise across various steps. Data is collected from W4A4 LDM-8 on LSUN-Churches.

Figure B: Comparison of the variance schedule and the scaled variance of quantization noise from LDM-4 on LSUN-Churches. Here, the scale for $\sigma_q$ refers to the coefficient in Eq. (12) in the paper.

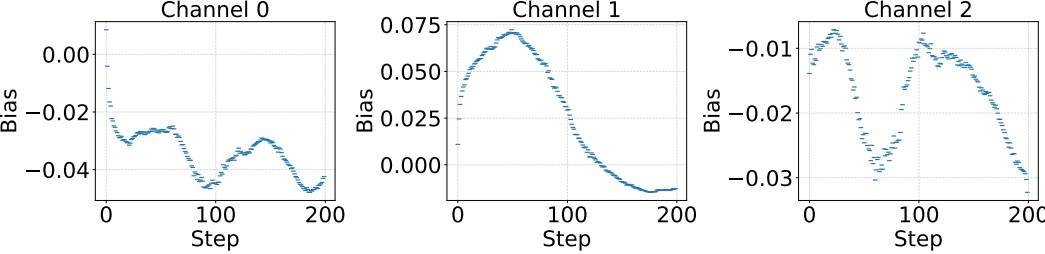

Figure C: Channel-wise bias of residual quantization noise. Data is collected from W4A4 LDM-8 on LSUN-Churches.

**Correlation analysis.** In Figures D to G, we present the results of linear regression analysis conducted on the quantization noise and the output of the full-precision noise estimation network, which includes Pearson's coefficient $R$ and the coefficient $k$ as defined in Eq. (B). As depicted in Figures E and G, we observe a notably high $R$ value for diffusion models with W4A4 bitwidth, indicating that the

quantization noise primarily consists of the correlated component. This finding demonstrates the effectiveness of our method in rectifying this specific aspect of quantization noise, particularly in scenarios involving low bitwidth. In cases of diffusion models with W4A8 or W8A8 bitwidth, our approach can also correct a substantial portion of the quantization noise by leveraging the correlation. Additionally, for larger steps, the coefficient $k$ generally exhibits positive values (which can also be observed in Figures E and G, as $k$ and $R$ value share the same sign), thereby affirming our capability to correct the correlated part of the quantization noise in these steps.

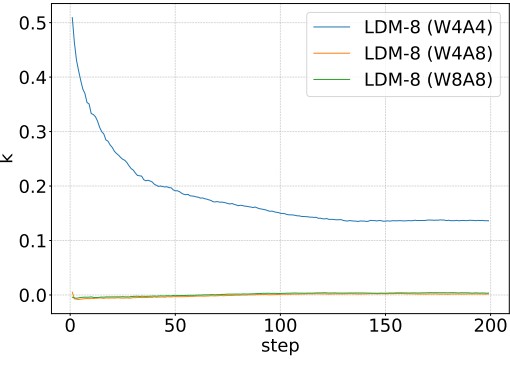

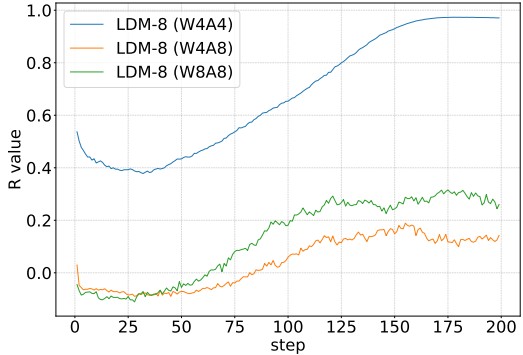

Figure D: The correlation coefficient $k$ in each step of LDM-8 on LSUN-Churches.

Figure E: The $R$ value of linear regression in each step of LDM-8 on LSUN-Churches.

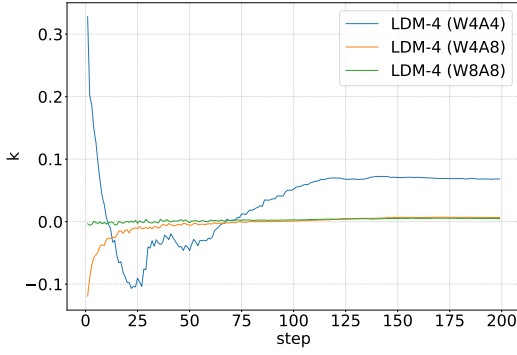

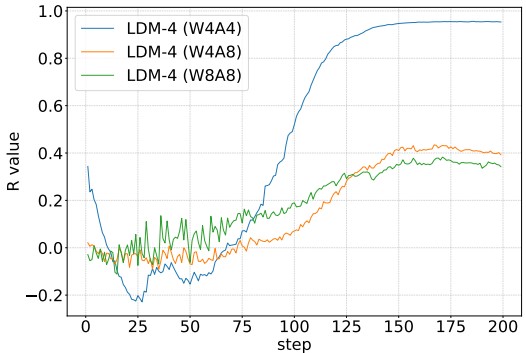

Figure F: The correlation coefficient $k$ in each step of LDM-4 on LSUN-Bedrooms.

Figure G: The $R$ value of linear regression in each step of LDM-4 on LSUN-Bedrooms.

## C   Additional experimental results

### C.1   Implementation details of step-aware mixed precision

Table A presents the results of bitwidth allocation for each dataset, which are determined by Eq. (15) in the paper.

Table A: Bitwidth allocation for each dataset.

| Dataset | W4A4 Step Range | W4A8 Step Range |
| --- | --- | --- |
| ImageNet (250 steps) | 249 to 202 | 201 to 0 |
| ImageNet (20 steps) | 19 to 15 | 14 to 0 |
| LSUN-Bedrooms | 199 to 155 | 154 to 0 |
| LSUN-Churches | 199 to 146 | 145 to 0 |

## C.2 Additional ablation experiments

In this section, we conduct additional ablation experiments with constant precision, which are outlined in Table B. The experimental results consistently demonstrate performance improvements brought by each component of our method under constant precision settings. Notably, our method exhibits more significant improvements at lower bitwidth (W3A8) due to the inherent presence of greater quantization noise at these levels.

Table B: Additional ablation study with constant precision on LSUN-Bedrooms dataset. As the bitwidth decreases, the efficacy of our approach becomes increasingly pronounced.

| Model | Method | Bitwidth (W/A) | FID↓ | sFID↓ |
|---|---|---|---|---|
| | FP | 32/32 | 3.00 | 7.13 |
| | Q-Diffusion | 4/8 | 6.72 | 18.80 |
| | +CNC | 4/8 | 6.31 | 16.28 |
| LDM-4 | +CNC+VSC | 4/8 | 6.10 | 16.03 |
| (steps=200 | PTQD (CNC+VSC+BC) | 4/8 | **5.94** | **15.16** |
| eta=1.0) | Q-Diffusion | 3/8 | 8.31 | 21.06 |
| | +CNC | 3/8 | 7.01 | 18.32 |
| | +CNC+VSC | 3/8 | 6.66 | 17.99 |
| | PTQD (CNC+VSC+BC) | 3/8 | **6.46** | **17.04** |

## C.3 Comparisons with PTQ4DM

Additionally, we include a comparison with the PTQ method PTQ4DM [9] on the LSUN-Bedrooms dataset, as shown in Table C. Remarkably, our proposed approach outperforms PTQ4DM in both W4A8 and W3A8 bitwidth scenarios.

Table C: Performance comparisons with PTQ4DM on LSUN-Bedrooms dataset over LDM-4 model.

| Model | Method | Bitwidth (W/A) | FID↓ | sFID↓ |
|---|---|---|---|---|
| | FP | 32/32 | 3.00 | 7.13 |
| | PTQ4DM | 4/8 | 20.72 | 54.30 |
| LDM-4 | Q-Diffusion | 4/8 | 6.72 | 18.80 |
| (steps=200 | Ours | 4/8 | **5.94** | **15.16** |
| eta=1.0) | PTQ4DM | 3/8 | 22.17 | 51.93 |
| | Q-Diffusion | 3/8 | 8.31 | 21.06 |
| | Ours | 3/8 | **6.46** | **17.04** |

## C.4 Evaluation with advanced sampler

Table D presents the results on a new dataset CelebA-HQ over recent DDPM variants PLMS [6], demonstrating the strong performance of PTQD under this configuration. Notably, the proposed PTQD reduces the FID and sFID by a considerable margin of 3.23 and 4.73 in comparison to Q-Diffusion, respectively.

Table D: Experimental results on CelebA-HQ dataset with PLMS sampler.

| Model | Method | Bitwidth (W/A) | FID↓ | sFID↓ |
|---|---|---|---|---|
| LDM-4 | FP | 32/32 | 16.72 | 15.97 |
| (steps=200 | Q-Diffusion | 4/8 | 24.31 | 22.11 |
| eta=0.0) | Ours | 4/8 | **21.08** | **17.38** |

Additionally, we present the results of our PTQD over latest DPM++ solver [7] on LSUN-Churches dataset, as shown in Table E. Notably, our PTQD with W3A8 bitwidth achieves a sFID result comparable to that of W4A8 Q-Diffusion.

Table E: Experimental results on LSUN-Churches dataset with DPM++ sampler.

| Model | Method | Bitwidth (W/A) | FID | sFID |
|---|---|---|---|---|
| | FP | 32/32 | 5.97 | 21.50 |
| LDM-8 | Q-Diffusion | 4/8 | 7.80 | 23.24 |
| (steps = 50 | Ours | 4/8 | **7.45** | **22.74** |
| eta = 0.0) | Q-Diffusion | 3/8 | 11.44 | 24.67 |
| | Ours | 3/8 | **10.72** | **23.36** |

## C.5 Evaluation with different variance schedule

Table F presents experimental results with deterministic and stochastic sampling on FFHQ and ImageNet dataset over LDM-4 model. While deterministic sampling has gained widespread adoption, it tends to result in lower output quality compared to stochastic sampling [11, 4]. Specifically, when generating samples on FFHQ dataset with a deterministic DDIM sampler, introducing stochastic perturbations lower both the FID and sFID metrics. For experiments on ImageNet dataset, it greatly improves the IS with little increase in FID and sFID.

Table F: Comparisons of generated sample quality under different variance schedules (denoted by *eta* below).

| Model | Dataset | *eta* | IS↑ | FID↓ | sFID↓ |
|---|---|---|---|---|---|
| LDM-4 | FFHQ | 0.0 | - | 11.26 | 8.36 |
| (steps=200) | | 1.0 | - | **9.37** | **7.04** |
| LDM-4 | ImageNet | 0.0 | 150.04 | **4.35** | **6.20** |
| (steps=250) | | 1.0 | **185.04** | 5.05 | 7.10 |

# D Additional visualization results

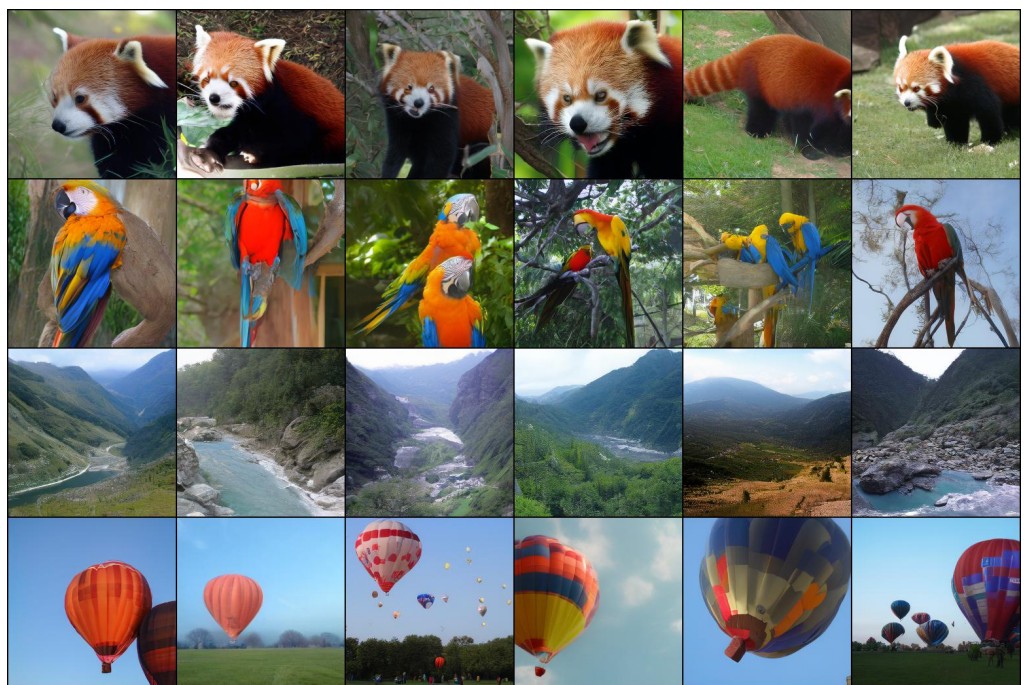

(a) samples generated with 20 steps

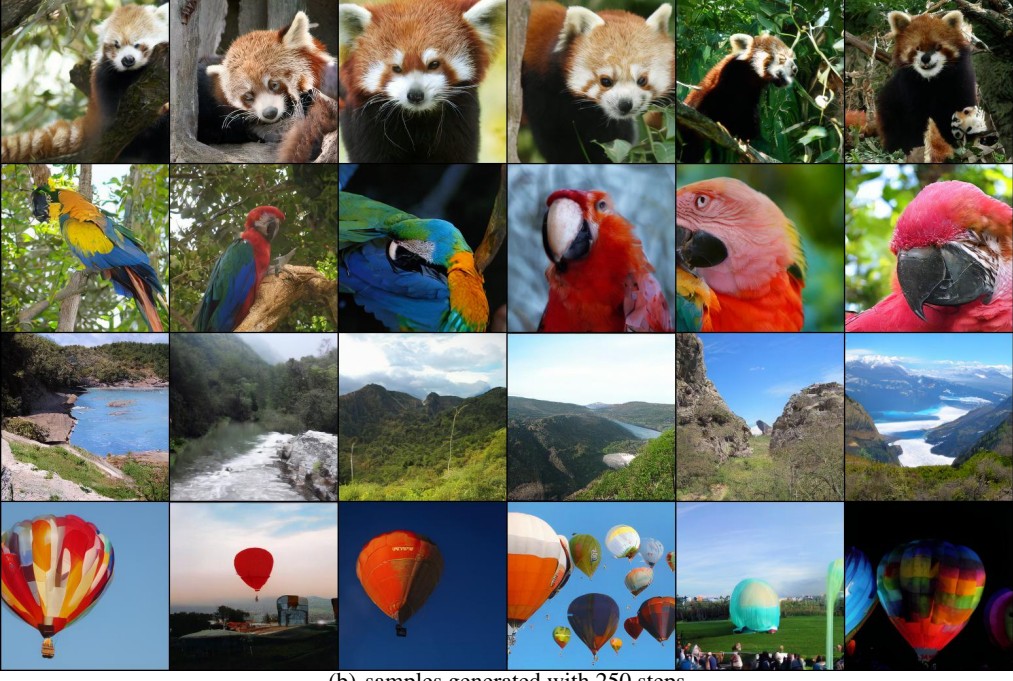

(b) samples generated with 250 steps

Figure H: Class-conditional generation on ImageNet $256 \times 256$. With the proposed PTQD, LDM-4 [8] with W4A8 bitwidth can generate high-fidelity images in only 20 steps.

| Q-Diffusion (MP) | PTQD (MP) | Q-Diffusion (W4A8) | PTQD (W4A8) | FP |
|---|---|---|---|---|

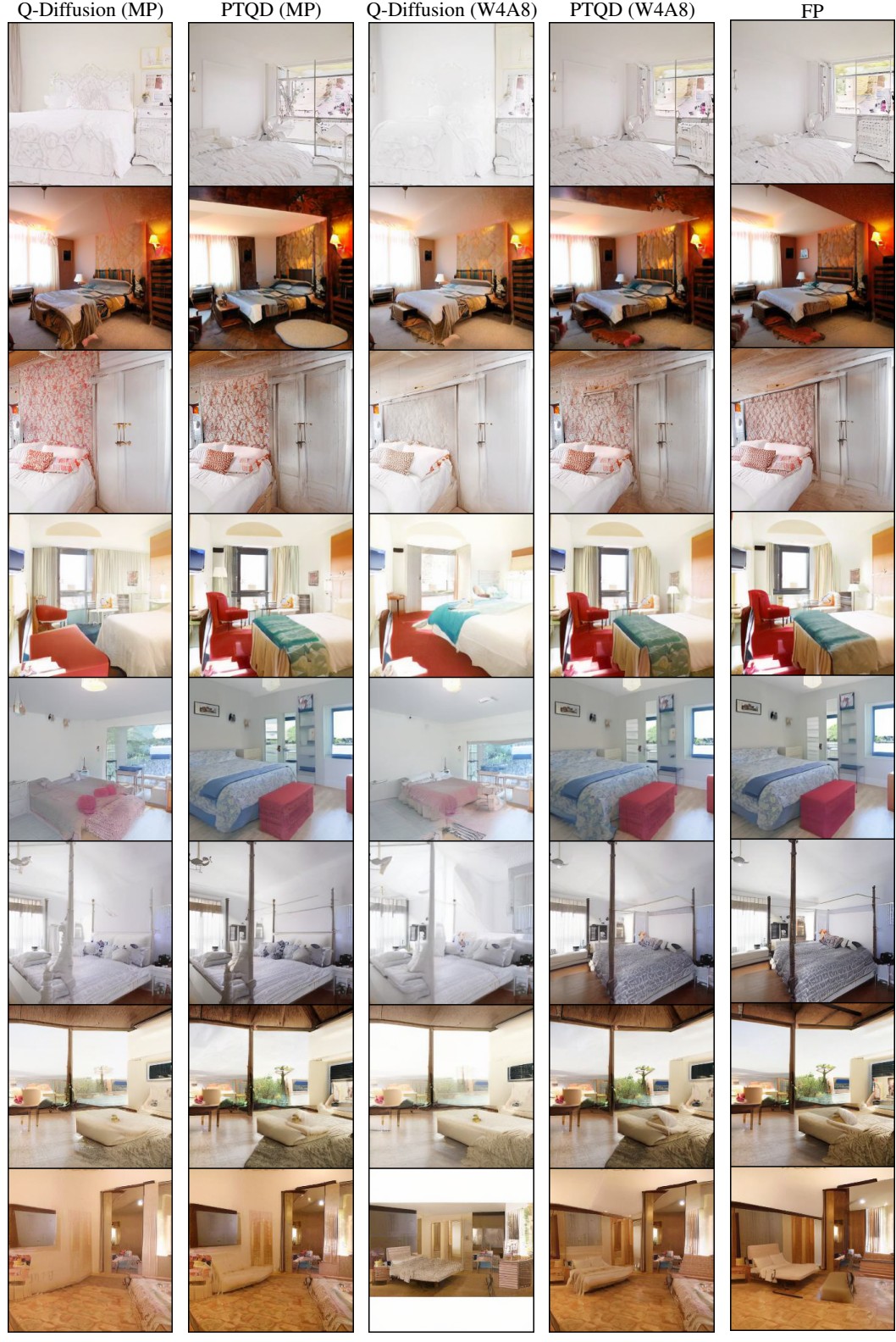

Figure I: The comparisons of samples generated by Q-Diffusion [5], PTQD and full-precision LDM-4 [8] on LSUN-Bedrooms $256 \times 256$. Compared with Q-Diffusion, samples generated by PTQD are less affected by quantization noise and exhibit a closer resemblance to the results of the full-precision model.

| Q-Diffusion (MP) | PTQD (MP) | Q-Diffusion (W4A8) | PTQD (W4A8) | FP |
|---|---|---|---|---|

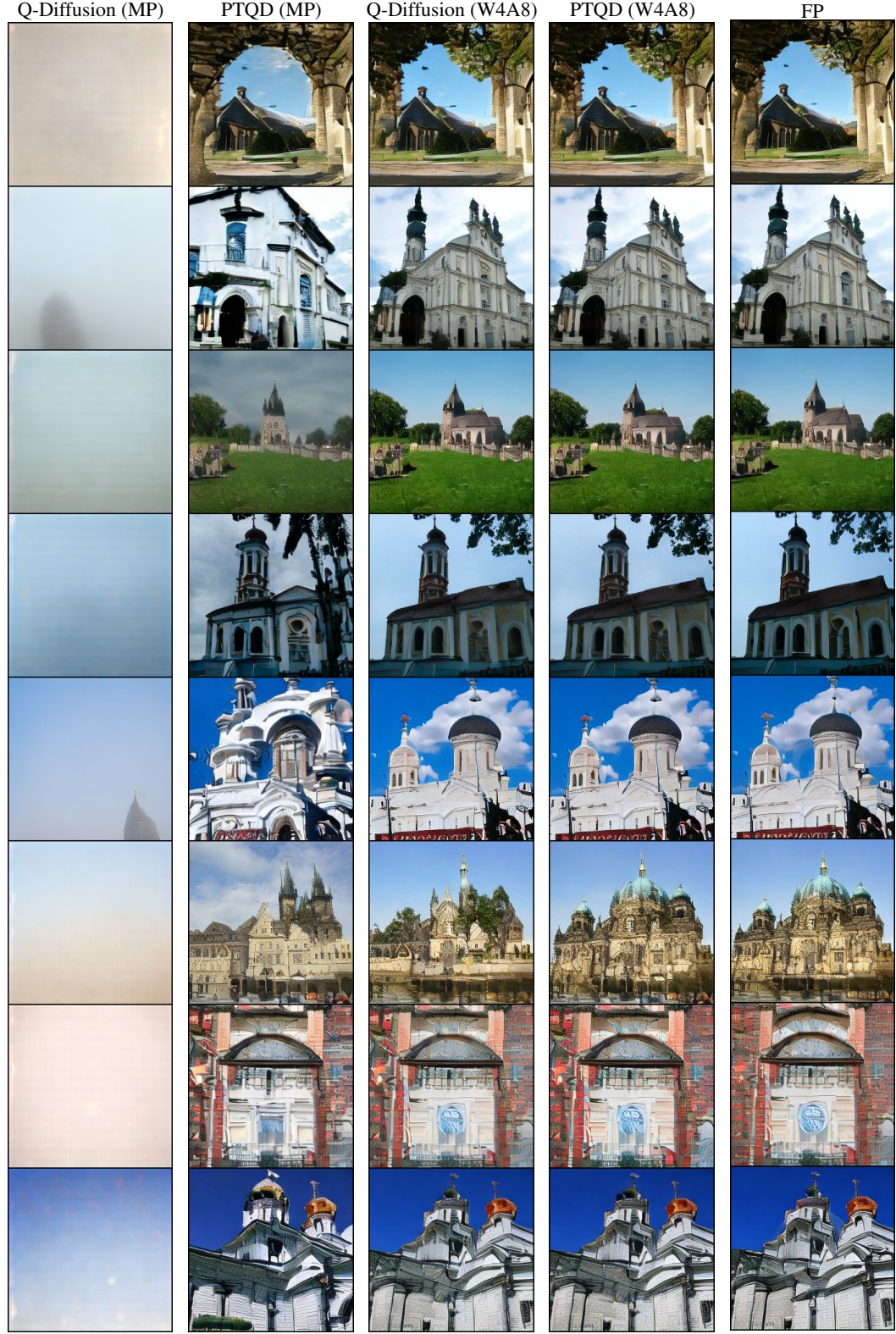

Figure J: The comparisons of samples generated by Q-Diffusion [5], PTQD and full-precision LDM-8 [8] on LSUN-Churches 256 × 256. While Q-Diffusion fails to denoise when utilizing W4A4 model in the mixed precision setting, PTQD, on the other hand, can still generate high-quality images under these conditions.