# OpenReview forum: "PTQD: Accurate Post-Training Quantization for Diffusion Models"
_NeurIPS.cc/2023/Conference — NeurIPS 2023 poster_

### Official Review · Reviewer_5dgo · 2023-06-20

**Soundness:** 3 good
**Presentation:** 3 good
**Contribution:** 3 good
**Rating:** 5
**Confidence:** 4

**Summary:**

The paper introduces PTQD, a Post-Training Quantization framework for Diffusion models.
PTQD analyzes the influence of quantization noise on diffusion noise.
The method suggests separating the quantization noise into noise correlated and uncorrelated with the full-precision reverse diffusion.
The correlated part is easily fixed by estimating the correlation coefficient.
The residual noise is corrected by modifying the stochastic variance of DM in reverse diffusion.
Finally, a mixed precision which is time-step aware is proposed based on gathered statistics.

**Strengths:**

The paper is clearly written.\
The analysis of DM quantization challenges is interesting, well-described, and well-mathematically defined and analyzed.\
The framework while being very simple (mainly based on gathered statistics) still can outperform previous work.

**Weaknesses:**

The main weaknesses are as follows:

1) The practical ability of the method to deal with the residual noise. (see question 1))

2) The weak comparisons. (see questions 3-4))


**Questions:**

1) As far as I know most DDPMs make no use of the stochastic reverse diffusion ($\sigma_t = 0$). In that case, the contribution and impact are certainly reduced.

2) A visualization of the generated images should be provided in order to ass the impact on both the generation and reconstruction quality.

3) Comparison with regular PTQ should be performed at different bit ranges (not only naive TensorRT's uniform quantization but others such as MMSE etc.)

4) Ablations: The ablation is good but should also be performed at a constant precision (not MP) in order to assess the real impact of each suggestion, independently of the mixed precision correction.

**Limitations:**

Limitations are ok.

---

> ### Author Rebuttal · Authors · 2023-08-09
>
> Thanks to the reviewer for the valuable comments.
>
> **Q1: Contribution and impact in deterministic sampling.** 1) We acknowledge that the efficacy of our contributions encounters constraints for deterministic sampling, as pointed out in line 205 of the paper. However, in the deterministic case, we can still **correct the correlated quantization noise** and the **biases stemming from uncorrelated components**. This corrective capability assumes particular significance in instances involving low bit quantization, as referred to the experimental result of LSUN-Churches in Table 3 in the paper and Figure J in supplementary material. 2) While deterministic sampling has gained widespread adoption, it tends to **result in lower output quality** compared to stochastic sampling [i, ii]. This proposition is also substantiated by empirical observations, as referred to Table C in the rebuttal PDF. Specifically, when generating samples on FFHQ dataset with deterministic DDIM sampler, introducing stochastic perturbations lower both the FID and sFID metrics. For experiments on ImageNet dataset, it greatly improves the IS with little increase in FID and sFID. In the case of stochastic sampling, our method can achieve better performance by calibrating the variance schedule.
>
> **Q2: A visualization of the generated images should be provided.** As referred to Figures H, I, J in the supplementary material, we have provided visualization results on three datasets to substantiate the effects of our proposed method.
>
> **Q3: Comparison with regular PTQ should be performed at different bit ranges.** As referred to Tables A and E in the rebuttal PDF, we have conducted experiments on more bitwidth settings and compared the results with PTQ4DM. Due to the limited time slot of the rebuttal, we will conduct experiments on more PTQ methods and bitwidths and add the results to the revised version.
>
> **Q4: The ablation should also be performed at a constant precision.** As referred to lines 219-220 in the paper, the proposed mixed-precision (MP) scheme allows the utilization of low-bit diffusion models during the sampling process, resulting in a greater speedup in generation. Specifically, we introduce W4A4 in the MP experiment, a **more intricate task** in comparison to the fixed W4A8 quantization due to the larger quantization noise. Moreover, we conduct additional ablation experiments with constant precision, which are outlined in Table A of the attached rebuttal PDF. The experimental results consistently demonstrate performance improvements brought by each component of our method under constant precision settings. Notably, our method exhibits more significant improvements at lower bitwidths (W3A8) due to the inherent presence of greater quantization noise at these levels.
>
> [i] Karras, Tero, et al. "Elucidating the design space of diffusion-based generative models." NeurIPS 2022.
>
> [ii] Song, Yang, et al. "Score-based generative modeling through stochastic differential equations." ICLR 2021.

---

### Official Review · Reviewer_MVe5 · 2023-06-30

**Soundness:** 2 fair
**Presentation:** 1 poor
**Contribution:** 2 fair
**Rating:** 5
**Confidence:** 4

**Summary:**

The paper suggests a method for post-training quantization of diffusion models. The method consists of a factorization of the quantization noise into a correlated and an uncorrelated part, and then addressing each component separately, either by linearly regressing for the correlation coefficient, or incorporating the quantization noise into the diffusion noise level. The authors conduct extensive experiments on multiple datasets, and obtain generation quality comparable to that of the full-precision model in most cases.

**Strengths:**

- The characterization of the quantization noise, and incorporating its uncorrelated part into the diffusion noise is a nice idea.
- The obtained results are impressive.

**Weaknesses:**

- Related work lacks a specific description of methodical/result-based differences between this work and Q-Diffusion. "Analyze the quantization effect" and "unified framework" are very broad terms that give little to no context.
- Lines 170-171: When does this linear regression happen? At training time or test time? What data do you use for this? Details lacking. Edit: Data is provided at the end of page 7, just please mention that details will be presented later from lines 170-171.
- What is $SNR^F$? What is the motivation behind it? Why is it defined this way? What is its purpose? These details should not be delegated to a citation.
- The reason not to compare with PTQ4DM sounds unreasonable. We can still compare results even if they do not specify the PTQ method used. Moreover, it seems like PTQ4DM has a public git repo with their code. The PTQ method can be inferred from that. Moreover, Q-Diffusion seems to have a curious failure mode for multi-precision. When considering a single precision level, it seems like the new method is not very different from Q-Diffusion in terms of results.
- No wall clock time comparison is given, even though slow runtime of diffusion models is touted as one of the main drawbacks of these models.

**Questions:**

- What does Figure 2 plot? Axis titles represent vectors. If the numbers plotted are entries in these vectors, it should be noted in the figure caption or in the text.
- Equation 12 and its following text gloss over the case "otherwise". Mathematically, $\sigma_t^2 = 0$ is *not* a solution for Eq. 11 in this case, and should not be presented as such. When quantization noise becomes larger than $\sigma_t^2$, especially in the mentioned deterministic case, the proposed method cannot deal with the noise. If it does, a thorough explanation is needed here.

**Limitations:**

Potential negative impact: Yes.
Limitations: No.

---

> ### Author Rebuttal · Authors · 2023-08-09
>
> Thanks to the reviewer for the valuable comments.
>
> **Q1: Methodical/result-based differences between this work and Q-Diffusion.** In terms of methodology, Q-Diffusion designed a calibration data collection method and applied the PTQ method BRECQ [i] to the diffusion model. In sharp contrast, we introduce a unified formulation for quantization noise and diffusion perturbed noise, as referred to Eq. (5) in the paper. We argue that quantization noise alters the mean and variance of the predicted noise in each sampling step, resulting in poor sample quality of the quantized diffusion model. Our approach corrects both correlated and residual quantization noises at every step to mitigate these adverse effects. It can be seamlessly integrated with Q-Diffusion or **any other PTQ method** to consistently enhance their performance.
>
> **Q2: Mention the details of linear regression in lines 170-171.** Linear regression is performed before test time. As referred to lines 258-262 of the paper, we generate 1024 samples using both quantized and full-precision diffusion models to collect the data (the quantization noise and the output of the full-precision noise prediction network) for performing linear regression. We will add the details in front in the revised version.
>
> **Q3: Details of $\rm{SNR}^F$ should not be delegated to a citation.** $\rm{SNR}$ is a widely adopted notation of signal-to-noise ratio. As referred to lines 226-229 in the paper and Eqs. (1)-(2) in [i], $\rm{SNR}^F$ is defined by the hyperparameters of the diffusion forward process, where $\alpha$ is the coefficient for data and $\sigma$ is the coefficient for noise. It is first introduced by [i] to note the degree of noise of data at each step. Moreover, as referred to lines 212-213 in the paper, $\rm{SNR}^Q$ is defined as the SNR of the quantized noise prediction network. We compare these two metrics to select the optimal bitwidth that satisfies the $\rm{SNR}$ requirement for effective denoising.
>
> **Q4: The reason not to compare with PTQ4DM sounds unreasonable.** Our initial attention was directed towards PTQ4DM [ii] upon its initial publication on arXiv, which did not release its code. The CVPR camera-ready paper was made public after the NeurIPS submission deadline. Additionally, PTQ4DM's experimental scope was confined to lower-resolution datasets, whereas our study encompasses datasets with higher resolutions. Nonetheless, we evaluate PTQ4DM on LSUN-Bedrooms dataset as shown in Table E in the rebuttal PDF. Our method outperforms it under both W4A8 and W3A8 bitwidths. Full results will be included in the revised version.
>
> **Q5: Q-Diffusion seems to have a curious failure mode for multi-precision.** This can be attributed to the substantial quantization noise inherent in the W4A4 bitwidth. In the absence of our correction method, the excessive noise becomes a hindrance, preventing Q-Diffusion from generating samples of desirable quality.
>
> **Q6: Method is not very different from Q-Diffusion in terms of results.** 1) It is essential to consider that the absolute performance **improvement is closely related to the precision** of the model. When higher bitwidths are employed, the absolute performance gains may appear relatively small, because the model's performance is already in close proximity to the full-precision counterpart. **As the bitwidth decreases, the efficacy of our approach becomes more noticeable**, particularly in the scenarios where W4A4 bitwidth is utilized, as referred to the results of mixed precision in Tables 1-3 in the paper. Notably, our method substantially reduces the FID score from 218.59 to 17.99 on the LSUN-Churches dataset. In addition, we conducted experiments using lower bitwidth W3A8 to demonstrate the extent of our improvement, as shown in Table A in the attached rebuttal PDF. The experimental results show that our method can bring greater improvement at W3A8 bitwidth on LSUN-Bedrooms dataset, resulting in a noticeable reduction of $1.85$ and $4.02$ in FID and sFID, respectively.
>
> 2\) It is worth noting that while FID provides an informative metric, it might not **holistically capture the improved image quality**. In the supplementary material, we have provided visualizations in Figures H-J that convincingly showcase the superiority of results produced by PTQD. These visualizations underscore higher image quality and a closer resemblance to samples generated by the full-precision model, in stark contrast to Q-Diffusion outputs.
>
> **Q7: No wall clock time comparison is given.** Please refer to Q3 in the general response.
>
> **Q8: What does Figure 2 plot?** Figure 2 illustrates the correlation between the quantization noise (Y-axis) and the output of the full-precision noise prediction network (X-axis). Each data point on the plot corresponds to specific entries within these vectors. We will clarify this point in the figure caption of the revised version.
>
> **Q9: The proposed method cannot deal with the noise in the deterministic case.** Please refer to Q4 in the general response.
>
> **Q10: Mathematically, Equation 12 is not a solution for Equation 11.** When quantization noise becomes larger than ${\sigma_{t}^{2}}$, ${\sigma_{t}^{'2}}=0$ is not an analytical solution of Eq. (11), but an optimal solution in this case. We will revise it in the revised version.
>
> [i] Kingma, Diederik, et al. "Variational diffusion models." NeurIPS 2021.
>
> [ii] Shang, Yuzhang, et al. "Post-training quantization on diffusion models." CVPR 2023.

---

> > ### Comment · Reviewer_MVe5 · 2023-08-12
> >
> > Thank you for your response. The added results and explanations following my fellow reviewers' and my suggestions indeed enrich the paper.
> >
> > Therefore, I change my recommendation to acceptance.

---

> > > ### Author Response · Authors · 2023-08-12
> > >
> > > Dear Reviewer MVe5,
> > >
> > > Thank you for your feedback. We truly appreciate your careful consideration of our responses to your and the other reviewers' suggestions.
> > >
> > > Best regards,
> > >
> > > Authors of #2641.

---

### Official Review · Reviewer_oS4i · 2023-07-04

**Soundness:** 3 good
**Presentation:** 3 good
**Contribution:** 3 good
**Rating:** 6
**Confidence:** 2

**Summary:**

The authors propose a new post-training quantization method for diffusion models titled PTQD that disentangles the quantization noise into correlated and uncorrelated parts regarding its full-precision counterpart, and demonstrate that PTQD generates as much high-quality samples as its full-precision counterpart.

**Strengths:**

(1) The idea of disentangling the quantization noise into correlated and uncorrelated parts regarding its full precision counterpart seems to be novel.

(2) PTQD shows better performance than previous methods in both class-conditional image generation and unconditional image generation.

**Weaknesses:**

(1) Although PTQD reduces BOPs greatly, it is doubtful whether the speed of PTQD generating samples is really faster than that of its full-precision counterpart in real time.

**Questions:**

N/A

---

> ### Author Rebuttal · Authors · 2023-08-09
>
> Thanks to the reviewer for the valuable comments.
>
> **Q1: The speed of PTQD in real time.** We have measured the latency of matrix multiplication and convolution operations in quantized and full-precision diffusion models using an RTX3090 GPU, as presented below. Both floating-point and quantized operations are implemented with CUTLASS. When both weights and activations are quantized to 8-bit, we observe a **2.03$\times$** reduction in latency compared to its full-precision counterpart over LDM-4. Moreover, when weights and activations are quantized to 4-bit, the speedup further increases to **3.34$\times$**. The mixed-precision settings explored in our experiments strike a good balance between latency and model performance.
>
> **Comparisons of time cost across various bitwidth configurations on ImageNet 256$\times$256.** Due to the current lack of a fast implementation for W4A8, we implement MP scheme with W8A8 and W4A4 kernels.
>
> | Model              | Bitwidth (W/A) | Model Size (MB) | FID   | sFID  | Time (s) |
> |--------------------|----------------|------------------|-------|-------|----------|
> | LDM-4 (steps=250, eta=1.0, scale=1.5) | 32/32 | 1603.35 | 5.05 | 7.10 | 5.46     |
> |                    | 8/8            | 430.06           | 4.02 | 5.81 | 2.68     |
> |                    | MP             | 234.51           | 6.44 | 8.43 | 2.45     |
> |                    | 4/4            | 234.51           | -    | -    | 1.63     |

---

> > ### Comment · Reviewer_oS4i · 2023-08-19
> >
> > Thanks for your response. I keep my original score.

---

### Official Review · Reviewer_LoUy · 2023-07-06

**Soundness:** 3 good
**Presentation:** 2 fair
**Contribution:** 2 fair
**Rating:** 5
**Confidence:** 5

**Summary:**

The paper proposed the quantization scheme for Diffusion models, where they disentangled the quantization noise into the correlated and uncorrelated parts;
Then, they incorporated the correlated part into diffusion-perturbed noise and calibrated the denoising variance schedule to absorb additional variance into diffusion noise.
They also suggested a mixed-precision quantization scheme for handling the difference in activations variance for each time step.

**Strengths:**

It is a novel point of view in incorporating quantization noise into diffusion noise.
It also sounds good to me to utilize the bias correction proposed in DFQ to handle the uncorrelated part.

**Weaknesses:**

It does not seem to show noticeable improvement compared to existing works, Q-diffusion.
It seems to require more experiments to show their performance. The ablation study was performed only when quantizing with mixed precision.
The mixed-precision scheme has any special things; the effect seems to be insignificant according to table 2.

**Questions:**

1.	It would be helpful if you the match the technique the author proposed with the description of the method sections. The terms of CNC, VSC and BC was suddenly advent in the result section.
2.	It would make it easy to follow your work if you provide your algorithm in the manuscript.
3.	could you provide how many bit-width is allocated for each step for a specific model.
4.	could you provide the result when setting k as arbitrarily?
5.	Why did you do the ablation study with mixed precision?
6.	In your result, the mixed-precision scheme seems not well worked out for Q-diffusion. could you provide the reasons?
7.	In Section 4.1, did you want to explain that quantization noise is divided into the correlated part and unrelated part due to normalization layers? Or is it just an example?
8.	In line 180, what are the bias and additional variance? Please notate them with symbols for clarity

**Limitations:**

There seems to have still room for improving the readability and clarity in the manuscript.

---

> ### Author Rebuttal · Authors · 2023-08-09
>
> Thanks to the reviewer for the valuable comments.
>
> **Q1: It does not seem to show noticeable improvement compared to existing works.** 1) It is essential to consider that the absolute performance **improvement is closely related to the precision** of the model. When higher bitwidths are employed, the absolute performance gains may appear relatively small, because the model's performance is already in close proximity to the full-precision counterpart. **As the bitwidth decreases, the efficacy of our approach becomes more noticeable**, particularly in the scenarios where W4A4 bitwidth is utilized, as referred to the results of mixed precision in Tables 1-3 in the paper. Notably, our method substantially reduces the FID score from 218.59 to 17.99 on the LSUN-Churches dataset. In addition, we conducted experiments using lower bitwidth W3A8 to demonstrate the extent of our improvement, as shown in Table A in the attached rebuttal PDF. The experimental results show that our method can bring greater improvement at W3A8 bitwidth on LSUN-Bedrooms dataset, resulting in a noticeable reduction of $1.85$ and $4.02$ in FID and sFID, respectively.
>
> 2\) It is worth noting that while FID provides an informative metric, it might not **holistically capture the improved image quality**. In the supplementary material, we have provided visualizations in Figures H-J that convincingly showcase the superiority of results produced by PTQD. These visualizations underscore higher image quality and a closer resemblance to samples generated by the full-precision model, in stark contrast to Q-Diffusion outputs.
>
> **Q2: Match the proposed techniques with the description of the method sections.** Thanks for the valuable comments. Correlated Noise Correction (CNC) is proposed in section 4.2.1 of the paper, which corrects the correlated part of the quantization noise. Both Bias Correction (BC) and Variance Schedule Calibration (VSC) are proposed in section 4.2.2 to correct the uncorrelated quantization noise, as referred to lines 194-201 of the paper. We will incorporate your suggestions in the revised version.
>
> **Q3: Provide the algorithm in the manuscript.** The algorithm is briefly summarized below and will be included in the revised version.
>
> **Before sampling:**
> | Algorithm Step | Description                                                                                         |
> |------|-----------------------------------------------------------------------------------------------------|
> | 1    | Quantize diffusion models with BRECQ [i] (or other PTQ method).                                   |
> | 2    | Generate samples with both quantized and FP model and collect quantization noise.                 |
> | 3    | Calculate the correlated coefficient $k$ based on Eq. (7), and the mean and variance of the residual quantization noise as per Eq. (10). |
>
> **For each sampling step:**
> | Algorithm Step | Description                                                                                         |
> |------|-----------------------------------------------------------------------------------------------------|
> | 4    | Correct the correlated part of the quantization noise by dividing the output of the noise prediction network by $1+k$. |
> | 5    | Calibrate the variance schedule by Eq. (12) and subtract the channel-wise biases from the output of quantized noise prediction network. |
>
> **Q4: How many bit-width is allocated for each step for a specific model?** As referred to lines 229-234 in the paper, the bitwidth for each step is determined by comparing $\rm{SNR}^Q$ with $\rm{SNR}^F$ as per Eq. (15). The results of bitwidth allocation for each dataset are presented below and will be included in the final version.
>
> | Dataset               | W4A4 Step Range | W4A8 Step Range |
> |-----------------------|-----------------|-----------------|
> | ImageNet (250 steps)  | 249 to 202     | 201 to 0        |
> | ImageNet (20 steps)   | 19 to 15       | 14 to 0         |
> | LSUN-Bedrooms         | 199 to 155     | 154 to 0        |
> | LSUN-Churches         | 199 to 146     | 145 to 0        |
>
>
> **Q5: Provide the result when setting $k$ arbitrarily.** As shown in the table below, setting $k$ arbitrarily can greatly impair the quality of generated samples. As referred to lines 168-171 and Eqs. (7)-(9) of the paper, if $k$ is set arbitrarily, the correction of the correlated quantization noise can be inaccurate and there will still be correlation between the remaining quantization noise and the output of the noise prediction network.
>
> | Model      | Method      | FID    | sFID   |
> |------------|-------------|--------|--------|
> | LDM-4      | Q-Diffusion | 6.72   | 18.80  |
> |            | Random $k$  | 18.98  | 46.52  |
> |            | Ours        | **5.94** | **15.16** |
>
> **Q6: Why ablation study is conducted with mixed precision?** Please refer to Q2 in the general response.
>
> **Q7: Why the mixed-precision scheme did not work well for Q-Diffusion?** This can be attributed to the substantial quantization noise inherent in the W4A4 bitwidth. In the absence of our correction method, the excessive noise becomes a hindrance, preventing Q-Diffusion from generating samples of desirable quality.
>
> **Q8: In Section 4.1, is the quantization noise divided into the correlated part and unrelated part due to normalization layers?** As referred to lines 149-163 of the paper, we prove that at least a portion of the correlation comes from the normalization layer. It is plausible that other nonlinear layers within the model could also contribute to this correlation.
>
> **Q9: In line 180, what are the bias and additional variance?** The bias and additional variance refer to the mean and variance of the residual quantization noise, which are denoted in Eq. (10) of the paper. We will make the descriptions consistent in the revised version.
>
> [i] Li, Yuhang, et al. "Brecq: Pushing the limit of post-training quantization by block reconstruction." ICLR 2021.

---

> ### Author Response · Authors · 2023-08-14
> **Follow-Up on Rebuttal**
>
> Dear Reviewer LoUy,
>
> We greatly appreciate the time and effort in reviewing our work. We have carefully considered your comments and suggestions and have made significant revisions to address the concerns you raised. We are eager to ensure that our paper meets the high standards of our respected reviewers.
>
> Please don’t hesitate to let us know if there is any additional feedback you might have at this stage.
>
> Best regards,
>
> Authors of #2641.

---

> > ### Comment · Reviewer_LoUy · 2023-08-15
> > **dpm++**
> >
> > Thank you for your effort to address my concern.
> > I have an additional suggestion.
> >
> > As in Q-Diffusion, could you provide the results of applying latest solver such as dpm++ into your quantized models ?

---

> > > ### Author Response · Authors · 2023-08-17
> > > **Response to dpm++**
> > >
> > > Dear Reviewer LoUy,
> > >
> > > Thanks for your kind suggestion. As referred to Table 4 in the rebuttal PDF and our response to Reviewer i5nq, we have conducted experiments over recent solver PLMS[i], demonstrating the strong performance of PTQD under this solver. Additionally, we present the results of our PTQD over latest DPM++[ii] solver on LSUN-Churches dataset, as shown below. Notably, our PTQD with W3A8 bitwidth achieves a sFID result comparable to that of W4A8 Q-Diffusion.
> > >
> > > | Model       | Method      | Bitwidth (W/A) | FID    | sFID   |
> > > |-------------|-------------|----------------|--------|--------|
> > > | LDM-8 (steps=50, eta=0.0)       | FP          | 32/32          | 5.97   | 21.50  |
> > > |             | Q-Diffusion | 4/8            | 7.80   | 23.24  |
> > > |             | Ours        | 4/8            | **7.45**   | **22.74**  |
> > > |             | Q-Diffusion | 3/8            | 11.44  | 24.67  |
> > > |             | Ours        | 3/8            | **10.72**  | **23.36**  |
> > >
> > >
> > >
> > > Once again, thank you for your time and commitment in reviewing our work.
> > >
> > > Best regards,
> > >
> > > Authors of #2641.
> > >
> > >
> > > [i] Liu, Luping, et al. "Pseudo numerical methods for diffusion models on manifolds." ICLR 2022.
> > >
> > > [ii] Lu, Cheng, et al. "Dpm-solver++: Fast solver for guided sampling of diffusion probabilistic models." arXiv 2022.

---

> > > > ### Comment · Reviewer_LoUy · 2023-08-18
> > > > **Thank you for your response.**
> > > >
> > > > Most of my concerns are addressed. So I'm willing to raise the score.

---

> > > > > ### Author Response · Authors · 2023-08-18
> > > > > **Thanks to your valuable feedback**
> > > > >
> > > > > Dear Reviewer LoUy,
> > > > >
> > > > > Thank you for your feedback. We truly appreciate your careful consideration of our responses.
> > > > >
> > > > > Best regards,
> > > > >
> > > > > Authors of #2641.

---

> > > > > ### Author Response · Authors · 2023-08-21
> > > > > **Kindly Inquiry to Reviewer LoUy**
> > > > >
> > > > > Dear Reviewer LoUy,
> > > > >
> > > > > We want to express our gratitude for your valuable feedback, and we are genuinely pleased to learn that most of your concerns have been addressed, and you are considering a score adjustment.
> > > > >
> > > > > We understand your busy schedule, but if you've determined the new score, we would be extremely grateful to know the outcome. If there are any additional concerns you have,  please feel free to let us know.
> > > > >
> > > > > Thank you once again for your time, guidance, and consideration.
> > > > >
> > > > > Best regards,
> > > > >
> > > > > Authors of #2641.

---

### Official Review · Reviewer_i5nq · 2023-07-27

**Soundness:** 3 good
**Presentation:** 3 good
**Contribution:** 3 good
**Rating:** 6
**Confidence:** 2

**Summary:**

This paper introduces PTQD, a novel method designed to tackle issues arising when applying existing post-training quantization techniques directly to low-bit diffusion models. The proposed approach disentangles quantization noise into its correlated and residual uncorrelated components regarding its full-precision counterpart, enabling separate correction for each part . Moreover, the authors present Step-aware Mixed Precision, a scheme that dynamically selects optimal bitwidths for individual denoising steps. Extensive experiments on three image datasets demonstrate significant improvements in image quality compared to the baseline.

**Strengths:**

1. PTQD presents a unique method that disentangles quantization noise and addresses it separately, while Step-aware Mixed Precision dynamically optimizes bitwidths for synonymous steps, demonstrating a comprehensive approach to quantization in diffusion models.
2. The experimental results provide strong evidence of PTQD's effectiveness in enhancing image quality, validating its practical value.
3. The paper is well-organized and clearly presented, with the supplementary file extending PTQD to DDIM and including a statistical analysis of residual quantization noise, which bolsters the credibility of their work.

**Weaknesses:**

1. To further validate PTQD's performance, it would be beneficial to include comparisons with more competitive methods, such as recent DDPM variants, in the experiments.
2. Evaluating PTQD with other post-training quantization methods on diverse image datasets would enhance its applicability and demonstrate its effectiveness across various tasks, making the findings more robust.

**Questions:**

See weakness.

**Limitations:**

None.

---

> ### Author Rebuttal · Authors · 2023-08-09
>
> Thanks to the reviewer for the valuable comments.
>
> **Q1: Include comparisons with recent DDPM variants and evaluating PTQD with other post-training quantization methods on diverse image datasets in the experiments.** Table D in the rebuttal PDF presents the results on a **new dataset CelebA-HQ** over recent **DDPM variants PLMS [i]**, demonstrating the strong performance of PTQD under this configuration. Notably, the proposed PTQD reduces the FID and sFID by a considerable margin of $3.23$ and $4.73$ in comparison to Q-Diffusion, respectively. Additionally, we include a comparison with the PTQ method PTQ4DM [ii] on the LSUN-Bedrooms dataset, as shown in Table E in the rebuttal PDF. Remarkably, our proposed approach outperforms PTQ4DM in both W4A8 and W3A8 bitwidth scenarios. The full results will be included in the revised version.
>
> [i] Liu, Luping, et al. "Pseudo numerical methods for diffusion models on manifolds." ICLR 2022.
>
> [ii] Shang, Yuzhang, et al. "Post-training quantization on diffusion models." CVPR 2023.

---

### Author Rebuttal · Authors · 2023-08-09

We thank all reviewers for their valuable feedback. Overall, our work has been well recognized as it "is well-organized and clearly presented" (Reviewer i5nq), presents a novel idea" (Reviewer oS4i) and "obtains impressive results" (Reviewer MVe5). We have summarized and addressed the main concerns as follows:

**Q1: Improvement is not noticeable compared to Q-Diffusion.** 1) It is essential to consider that the absolute performance **improvement is closely related to the precision** of the model. When higher bitwidths are employed, the absolute performance gains may appear relatively small, because the model's performance is already in close proximity to the full-precision counterpart. **As the bitwidth decreases, the efficacy of our approach becomes more noticeable**, particularly in the scenarios where W4A4 bitwidth is utilized, as referred to the results of mixed precision in Tables 1-3 in the paper. Notably, our method substantially reduces the FID score from 218.59 to 17.99 on the LSUN-Churches dataset. In addition, we conducted experiments using lower bitwidth W3A8 to demonstrate the extent of our improvement, as shown in Table A in the attached rebuttal PDF. The experimental results show that our method can bring greater improvement at W3A8 bitwidth on LSUN-Bedrooms dataset, resulting in a noticeable reduction of $1.85$ and $4.02$ in FID and sFID, respectively.

2\) It is worth noting that while FID provides an informative metric, it might not **holistically capture the improved image quality**. In the supplementary material, we have provided visualizations in Figures H-J that convincingly showcase the superiority of results produced by PTQD. These visualizations underscore higher image quality and a closer resemblance to samples generated by the full-precision model, in stark contrast to Q-Diffusion outputs.

**Q2: Why conduct ablation study with mixed precision?** As referred to lines 219-220 in the paper, the proposed mixed-precision (MP) scheme allows the utilization of low-bit diffusion models during the sampling process, resulting in a greater speedup in generation. Specifically, we introduce W4A4 in the MP experiment, a **more intricate task** in comparison to the fixed W4A8 quantization due to the larger quantization noise. Moreover, we conduct additional ablation experiments with constant precision, which are outlined in Table A of the attached rebuttal PDF. The experimental results consistently demonstrate performance improvements brought by each component of our method under constant precision settings. Notably, our method exhibits more significant improvements at lower bitwidths (W3A8) due to the inherent presence of greater quantization noise at these levels.

**Q3: Real time speed up of PTQD.** We have measured the latency of matrix multiplication and convolution operations in quantized and full-precision diffusion models using an RTX3090 GPU, as shown in Table B in the rebuttal PDF. Both floating-point and quantized operations are implemented with CUTLASS. When both weights and activations are quantized to 8-bit, we observe a **$2.03\times$** reduction in latency compared to its full-precision counterpart over LDM-4. Moreover, when weights and activations are quantized to 4-bit, the speedup further increases to **$3.34\times$**. The mixed-precision settings explored in our experiments strike a good balance between latency and model performance.

**Q4: The contributions are reduced for deterministic sampling ($\sigma_t=0$).** 1) We acknowledge that the efficacy of our contributions encounters constraints for deterministic sampling, as pointed out in line 205 of the paper. However, in the deterministic case, we can still **correct the correlated quantization noise** and the **biases stemming from uncorrelated components**. This corrective capability assumes particular significance in instances involving low bit quantization, as referred to the experimental result of LSUN-Churches in Table 3 in the paper and Figure J in supplementary material. 2) While deterministic sampling has gained widespread adoption, it tends to **result in lower output quality** compared to stochastic sampling [i, ii]. This proposition is also substantiated by empirical observations, as referred to Table C in the rebuttal PDF. Specifically, when generating samples on FFHQ dataset with deterministic DDIM sampler, introducing stochastic perturbations lower both the FID and sFID metrics. For experiments on ImageNet dataset, it greatly improves the IS with little increase in FID and sFID. In the case of stochastic sampling, our method can achieve better performance by calibrating the variance schedule.

[i] Karras, Tero, et al. "Elucidating the design space of diffusion-based generative models." NeurIPS 2022.

[ii] Song, Yang, et al. "Score-based generative modeling through stochastic differential equations." ICLR 2021.

---

### Decision · Program_Chairs · 2023-09-21

**Decision:**

Accept (poster)

**Comment:**

The paper proposes an interesting concept for diffusion models. The authors successfully answered the concerns of the reviewers and now all think the paper should be accepted. Therefore, I recommend accepting the paper. Clearly, the authors should improve the paper in the revision using the reviewers' comments.